



# Characterisation of Central-African aerosol and trace-gas emissions based on MAX-DOAS measurements and model simulations over Bujumbura, Burundi.

Clio Gielen[1], François Hendrick[1], Gaia Pinardi[1], Isabelle De Smedt[1], Caroline Fayt[1], Christian Hermans[1], Trissevgeni Stavrakou[1], Maite Bauwens[1], Jean-Francois Müller[1], Eugène Ndenzako[2], Pierre Nzohabonayo[2], Rachel Akimana[2], Sebastien Niyonzima[2], Michel Van Roozendael[1], and Martine De Mazière[1]

[1]Royal Belgian Institute for Space Aeronomy (BIRA-IASB), Brussels, Belgium
[2]Department of Physics, University of Burundi, Bujumbura, Burundi

*Correspondence to:* C. Gielen (clio.gielen@aeronomie.be)

**Abstract.**

We present MAX-DOAS measurements of $NO_2$, HCHO, and aerosols performed in Central Africa, in the city of Bujumbura, Burundi (3.38°S, 29.38°E). A MAX-DOAS instrument has been operated at this location by BIRA-IASB since late 2013. Aerosol-extinction and trace-gas vertical profiles are retrieved by applying the optimal-estimation-based profiling tool bePRO to the measured $O_4$, $NO_2$ and HCHO slant-column densities. The MAX-DOAS vertical columns and profiles are used for investigating the diurnal and seasonal cycles of $NO_2$, HCHO, and aerosols. Regarding the aerosols, the retrieved AODs are compared to co-located AERONET sun-photometer measurements for verification purposes, while in the case of $NO_2$ and HCHO, the MAX-DOAS vertical columns and profiles are compared to GOME-2 and OMI satellite observations.

To characterise the biomass-burning and biogenic emissions in the Bujumbura region, the trace gases and aerosol MAX-DOAS retrievals are used in combination with MODIS fire radiative-power values and the tropospheric 3D chemical transport model IMAGES, as well as simulations from the NOAA backward-trajectory model HYSPLIT. The first results show that the aerosol and HCHO seasonal variation is driven by the alternation of rain and dry periods, the latter being associated with intense biomass-burning agricultural activities and forest fires in the south/south-east and transport from this region to Bujumbura. In contrast, $NO_2$ is seen to depend mainly on local emissions close to the city, due to the short lifetime of this species (typically 1-2 hours).



## 1    Introduction

Central Africa is known for its strong biogenic, pyrogenic, and to a lesser extent, anthropogenic emissions, which release large amounts of aerosols and trace gases in the atmosphere that are transported across and between continents via long-range trajectories.

Satellite observations of species like nitrogen dioxide ($NO_2$) and formaldehyde (HCHO), in combination with inverse modelling tools, have pointed to large uncertainties associated with the emissions in this region (Marais et al., 2012; Miyazaki et al., 2012; Stavrakou et al., 2015; De Smedt et al., 2015). There is thus a need for additional measurements, especially from the ground, in order to better characterise the biomass-burning, anthropogenic, and biogenic emissions in this area.

Biomass-burning activities are commonplace in Tropical Africa and a major source of atmospheric pollutants that are re-
leased in the atmosphere (Crutzen and Andreae, 1990; Andreae and Merlet, 2001; Roberts et al., 2009; Akagi et al., 2011; Marais et al., 2012). These burning activities are mainly the clearing of woodland, shrubland, and cropland for agricultural use or charcoal production, and take place during the local dry seasons (Roberts et al., 2009). Anthropogenic emissions, such as aerosols and $NO_2$, are due to domestic heating (mainly fuel wood and charcoal) and industry and traffic (fossil fuels), and occur throughout the year, with little seasonality due to the low temperature variability in the Tropics.

In this study we use MAX-DOAS measurements to study the emission of aerosols, formaldehyde (HCHO), and nitrogen dioxide ($NO_2$) in the Central-African country of Burundi. All these species play an important role in the chemistry of the atmosphere and have a strong impact on local air quality. Formaldehyde is mainly produced by methane ($CH_4$) oxidation, but emissions of non-methane volatile organic compounds (NMVOCs) from biogenic, biomass-burning and anthropogenic sources can result in strong localised photochemical production resulting from their oxidation (Andreae and Merlet, 2001;
Seco et al., 2007; De Smedt et al., 2015). The global sink of HCHO is due to photolysis and oxidation by OH, resulting in a photochemical lifetime of only a few hours, making it an ideal tracer of NMVOC emission. Nitrogen dioxide is mainly released in the atmosphere due to human activities (the burning of fossil fuels), although some biogenic sources, such as lightning and soil activity, can also produce nitrogen oxides ($NO_x=NO+NO_2$). NO emitted from these sources is rapidly converted to $NO_2$ by a reaction with ozone ($O_3$) (Crutzen, 1970; Lee et al., 1997). Atmospheric aerosols are produced by a wide variety of
anthropogenic and natural sources. Primary particles are directly emitted from biomass burning, burning of fossil fuels, and volcanic eruptions. Secondary aerosols are formed in the atmosphere from gas-phase condensations and chemical reactions. $NO_2$ and VOCs are known precursors of these secondary aerosols. The lifetime of $NO_2$, which is mainly determined by the $OH + NO_2$ reaction during the daytime is typically only a few hours.

We present ground-based multi-axis differential absorption spectroscopy (MAX-DOAS) measurements of $NO_2$, HCHO,
and aerosols performed in the city of Bujumbura, Burundi (3.38°S, 29.38°E). A MAX-DOAS instrument has been operated at this location by BIRA-IASB since late 2013. The MAX-DOAS technique has been demonstrated to be ideally suited for the retrieval of tropospheric trace gases and deriving information on aerosol properties (e.g. Frieß et al., 2006; Clémer et al., 2010; Hendrick et al., 2014; Wang et al., 2014; Franco et al., 2015; Frieß et al., 2016; Schreier et al., 2016). A description of the measurement site of Bujumbura and local meteorological conditions is given in Section 2. An overview of data sets used





in the paper is given in Sect. 3, including a detailed description of the MAX-DOAS instrument, the DOAS technique and measurement retrieval scheme (Sect. 3.1), and the AERONET and MODIS data, as well as IMAGES and HYSPLIT models. The results of our analysis can be found in Sect. 4, where we investigate in more detail the observed seasonal and diurnal variations (Sect. 4.1) of the retrieved columns and profiles, and compare the MAX-DOAS results with additional data sets and

models to study the impact of local and regional emissions (Sect. 4.2). We end with general conclusions in Sect. 5

## 2    Site description

In the framework of the AGACC-II (Advanced exploitation of Ground-based measurements: Atmospheric Chemistry and Climate applications II, http://agacc.aeronomie.be) project, a MAX-DOAS instrument was installed in November 2013 on the building of the Department of Physics of the University of Burundi ($3.38°$S, $29.38°$E). The instrument is located at an

altitude of $860\,$m above sea level, at the eastern edge of the capital city of Bujumbura, with an approximated population of $500\,000$ inhabitants. The surrounding orography of the site is rather peculiar (Fig. 1): the city is located at the north-eastern tip of Lake Tanganyika, serving as the main port of the country. The lake and the city are situated in a valley, surrounded by mountainous walls of the Alberine Rift (the western expansion of the East-African Rift), reaching heights of $2000 - 3000\,$m. The north/north-west part of Bujumbura city consists of large plains with agricultural rice fields. The north-west is also the

location of the Rusizi park where there is the possibility of bush fires in June and August.

The region around Bujumbura has two rain seasons, a first one in March-June with a peak in April (with resulting green vegetation in the mountains), and a second one starting in September until December. There are also two dry seasons: the most important one is in July-August, the other one is January-February.

Several fire activities take place in the surroundings, at different times throughout the year: in September-October there is

agricultural waste burning for the preparation of the ground for the 'small' harvest season. In January-February-mid March there is again agricultural waste burning for the main harvest season. Large bush and forest fires occur in July-August in the east and south-east mountains. In these months the mornings also tend to be cooler, resulting in additional domestic heating in the morning in Bujumbura city.

### 2.1    Meteorological measurements

In January 2015 a small meteorological station was installed, to provide additional information on the temperature, humidity, wind speed, and wind direction at the site. The monthly and daily average meteorological values for 2015 can be found in Figure 2. Being an equatorial site, Bujumbura has a very stable yearly average temperature around $25°$C. The relative humidity is high, with typical values around $70\%$ and a yearly minimum at $50\%$ in August during the dry season. The wind speed and direction show a clear daily pattern, with winds increasing in speed up to $2\,$m/s before noon and decreasing in the afternoon,

with a direction change from around $120°$, thus coming from the east-south-east, in the morning and evening, and peaking at $240°$, coming from the west-south-west, at noon. It is worth noting that due to the location of the city in the valley, with





surrounding high mountain ranges, the winds at the site altitude could be markedly different from the winds prevalent at higher altitudes.

Due to the equatorial location of the site, Bujumbura does not experience the classical seasons of winter, spring, summer, and fall, so instead of using these to define different periods of the year, we will divide the year in four 3-month periods further

in the paper, i.e. DJF for December-January-February, MAM for March-April-May, JJA for June-July-August, and SON for September-October-November.

## 3   Description of the MAX-DOAS measurements and additional data sets

### 3.1   The MAX-DOAS instrument

The BIRA-IASB MAX-DOAS instrument is a fully-automated dual-channel system, consisting of a thermo-regulated box with

two spectrometers located inside the building. An optical head is mounted on a commercial EKO sun tracker located outside and connected to the spectrometers via optical fiber. The optical-head design is such that the telescope can be moved in a wide range of elevations $(0 - 90°)$, as well as azimuth directions $(0 - 360°)$. The instrument initially had a western viewing direction $(77°$ W), towards the city center and Lake Tanganyika. However, a systematic daily development of a cloud layer above the lake in the afternoon was observed, which we suspected hindered our measurements (see Sect. 4.1.3). It was therefore decided

to take additional measurements pointing away from the lake, towards the south, in alternation with the western measurements. These alternating west-south measurements started in August 2015.

The instrument is a passive DOAS instrument that performs quasi-simultaneous measurements of scattered sunlight for a range of different elevations, from the horizon to the zenith (Hönninger et al., 2004; Platt and Stutz, 2008). This results in an enhanced sensitivity to absorbing species in the lower troposphere compared to zenith observing techniques.

Information on aerosol characteristics, i.e. optical depth and extinction profile, is obtained using $O_4$ differential slant-column densities (DSCDs). This is possible as the vertical distribution of $O_4$ is well known and nearly constant, as it varies with the square of the $O_2$ monomer. The optical depth of $O_4$ is very sensitive to changes in the light path distribution and deviations of the $O_4$ DSCD from values representative for a clear sky are caused by aerosols or clouds. Measurements of the $O_4$ DSCD can therefore be used for the retrieval of aerosols (Hönninger et al., 2004; Wagner et al., 2004; Frieß et al., 2006).

At Bujumbura we have measurements from November 2013 until now, however, due to instrumental issues, we do not have continuous measurements for both UV and visible channels. For the UV channel, nearly a year of measurements (from 24/4/2014-13/4/2015) is missing. For the visible channel, almost 2 months of data are missing in May-July 2015 (20/5/2015-12/7/2015), as well as a month of missing measurements in January 2016.

### 3.1.1   DOAS data analysis

The MAX-DOAS spectra are analysed using the DOAS method (Platt and Stutz, 2008). This method separates narrow-band differential absorption patterns (which can be related to specific molecules in the atmosphere) from broad-band extinction





caused by Rayleigh and Mie scattering due to scattering on molecules and particles. This results in differential slant-column densities (DSCDs), i.e. the integrated concentration of an absorbing molecular species along the effective light path relative to the integrated concentration along the average light path of a reference spectrum. Here, the zenith spectrum of the scan is used as the reference spectrum. To analyse the MAX-DOAS spectra the spectral-fitting software package QDOAS is used (Dankaert et al., 2013).

A detailed overview of the DOAS settings for all the retrieved species can be found in Table 1.

### 3.1.2 Column and profile retrievals

To retrieve the aerosol and trace-gas characteristics, we use the bePRO radiative-transfer code (Clémer et al., 2010). This is an inversion algorithm based on the optimal estimation method (OEM, Rodgers, 2000) which uses the observed MAX-DOAS DSCDs to derive vertical profiles of the aerosol and trace-gas extinction at different wavelengths. The forward model in bePRO is the linearised discrete ordinate radiative transfer (LIDORT) model (Spurr, 2008). A detailed description of the bePRO algorithm and parameters is provided in Clémer et al. (2010) and Hendrick et al. (2014). It is found that the model retrievals are most sensitive to concentrations close to the surface, below $1\,km$, and typically contains about $1-2$ pieces of independent information (DFS, degrees of freedom for signal) for aerosol retrievals, and $2-3$ for trace-gas retrievals.

For Bujumbura we retrieve aerosol vertical profiles at 360 and $477\,nm$ from the $O_4$ DSCDs. These results give an indication of the light path through the atmosphere and are then used for the retrieval of the trace-gas profiles of $NO_2$ and HCHO at $460\,nm$ and $342\,nm$, respectively. The aerosol retrievals at 360 and $477\,nm$ are converted respectively to 342 and $460\,nm$ using the Ångström coefficient provided by AERONET co-located sun photometer measurements.

Important parameters in the OEM model are the a-priori profile and the weighting-function matrix. The latter expresses the sensitivity of the measured DSCDs to changes in the profile and it is calculated using the linearised discrete ordinate radiative-transfer model LIDORT (Spurr, 2008). The a-priori profile for the model is constructed as an exponentially-decreasing function of the form $\frac{COL_a}{SH}e^{-\frac{z}{SH}}$, with $z$ the altitude, SH the scaling height which we fix at a moderate value of $1.0\,km$. $COL_a$ is an a-priori value for the aerosol optical depth (AOD) or trace-gas vertical column density (VCD), which we fix at AOD$=0.5$ for the aerosols and derive from the geometrical approximation (Brinksma et al., 2008) at $30°$ elevation for the trace gasses. The pressure and temperature profiles are taken from the US Standard Atmosphere (1976) and the retrieval grid consists of eleven layers of $200\,m$ thickness between the station altitude and $3\,km$, $1\,km$ layers between 3 and $6\,km$, and one layer between 6 and $8\,km$. The albedo was fixed at 0.07.

Contrary to MAX-DOAS measurements in Xianghe (China) (Clémer et al., 2010), we do not find it necessary to apply a correction factor to the measured $O_4$ DSCDs. At this point it is unclear what the origin is of the sometimes observed discrepancy between the measured and modelled $O_4$ DSCDs for other measurement sites (e.g. Irie et al., 2015; Ortega et al., 2016).

Only those results where the retrieved DSCDs have a percent root mean square difference (RMS), defined as $\sqrt{\frac{\sum(measured-modelled)^2}{\sum measured^2}}$, smaller than $50\,\%$ are kept, since high RMS values typically point to a failure of the model during the retrieval process. We also discard retrievals with a DFS below 1 as they do not contain enough independent information. We furthermore perform



a cloud-screening process on our measurement spectra, following the method described in Gielen et al. (2014). For the cloud screening we use information on the colour index, derived from the intensity ratio at 410 and 530 nm, from both the zenith and 30° elevation angle. The method provides different flags denoting the sky conditions, such as clear-sky conditions, thin clouds or low-medium pollution, and thick clouds or extreme pollution events. The cloud screening furthermore provides a

flag for the presence of broken or scattered clouds. For this study we remove those measurements which are affected by thick and broken/scattered clouds. Unless stated otherwise, all results in this paper are cloud-screened measurements. As our cloud-screening method is unable to distinguish with certainty between optically thick clouds and extreme pollution events (AOD $\gg 2$), such events will also be removed from our measurements. However, such extreme pollution events are not expected to occur at Bujumbura.

An example of the model fit to the measured $O_4$ and trace-gas DSCDs can be found in Fig. 3. We find a good agreement between the bePRO model and the measurements, for both aerosol and trace-gas retrievals, with RMS values on average below 10 and 20% respectively. We find an average DFS for the model retrievals around 2. The resulting profile information for one measurement in this example day is presented in Fig. 4. The averaging kernels show that the bePRO retrievals are most sensitive close to the surface, in the first and second 200 m layer above the station. On average, the averaging kernels for the aerosols

show additional sensitivity to slightly higher layers, which is explained by the fact that for the trace gasses, the vertical profiles show a distribution which is more peaked at the lowest elevation layers (see Sect. 4.1.2). For both aerosols and trace gasses, we find that the retrieved profile generally follows the a-priori profile above $3 - 4$ km. The errors presented in Fig. 4 represent the difference between the retrieved profile and the true profile due to vertical smoothing by the algorithm (smoothing error) and the error due to the DOAS fit (noise error). It can be seen that the smoothing error is generally larger than the noise error,

with the exception of some retrievals in the lowest layer. However, we do note that the reported DOAS fit error is most likely a lower limit, as some systematic error sources (such as the Ring effect or the instrument slit function) are not taken into account when determining the DOAS fit error (Pinardi et al., 2013).

   In Table 2 we give an overview of the main error sources on the AOD and VCD retrievals. The uncertainty related to the choice of the a-priori profile (see above) is estimated by varying the adopted scale height between 0.5 and 1.0. This gives a

difference in AOD and VCD retrievals between $15 - 25\%$. The combined smoothing and noise error has values around 10% for the AOD and 15% for the trace-gas VCDs. The aerosol profile retrievals are used as input for the trace-gas retrievals, and thus introduce an additional source of uncertainty. To test the influence of varying aerosol profiles, we scaled the aerosol profiles to the measured AERONET AOD values. As we will show in Sect. 4.1.1 and Figure 10 specifically, the AERONET AOD values are often seen to reach higher values than the bePRO AOD retrievals. Scaling the retrieved bePRO profiles to the AERONET

AOD values, varies the retrieved trace-gas VCDs by about 5%. Combined with the systematic uncertainties on the $O_4$, $NO_2$ and HCHO cross-sections, 5%, 3%, and 9%, respectively, (Vandaele et al., 1998; Pinardi et al., 2013; Thalman and Volkamer, 2013), this gives total uncertainties on the AOD of 21%, on the $NO_2$ VCDs of 21%, and 31% for HCHO VCDs.





## 3.2  AERONET

Together with the MAX-DOAS instrument, a cimel sun-photometer was installed at the site, in the framework of AERONET (Aerosol Robotic Network, http://aeronet.gsfc.nasa.gov). The cimel is a sun-tracking instrument and thus does not continually point in the same direction as the MAX-DOAS. For the comparison with the MAX-DOAS retrieved AOD values, we used the
level 1.0 non-cloud-screened and level 1.5 cloud-screened AERONET data. We re-binned all the data to 0.2 h averages. We use the Ångstrom coefficients calculated by AERONET to convert the AERONET aerosol data, measured at 340, 380, 440, and 500 nm, to the MAX-DOAS wavelengths of 360 and 477 nm. A description of the comparison between AERONET and MAX-DOAS measurements is presented in Sect. 4.1.1.

## 3.3  MODIS

In Fig. 5 we show the monthly total fire radiative power for 2015, as measured by MODIS (Moderate Resolution Imaging Spectroradiometer), a closer look at the region around Bujumbura can be seen in Fig. 6. The MODIS data were taken from the MODFIRE collection 6 data products (http://feer.gsfc.nasa.gov/data/frp/), which provide an active fire product containing instantaneous fire radiative power (FRP) measurements at 1 km resolution from Terra and Aqua satellites. We can see that Central Africa is strongly affected by biomass-burning emissions. Most fire activities in the northern hemisphere occur in
December-January, whereas fire activities are predominant in the southern hemisphere during the middle of the year, with a peak around August. A more detailed comparison between the MODIS FRP and MAX-DOAS measurements is presented in Sect. 4.2.1.

## 3.4  IMAGES

The IMAGES v2 global chemical-transport model calculates the concentrations of 131 transported and 41 short-lived trace
gases with a time step of 6 hours at $2° \times 2.5°$ resolution between the surface and the lower stratosphere. Advection is driven by ERA-Interim meteorology obtained from the European Centre for Medium-range Weather Forecasts (ECMWF). Diurnal variations are accounted for through correction factors on the photolysis and kinetic rates obtained from a model simulation with a shorter (20 minute) time step. This simulation is used to calculate the diurnal shapes of $NO_2$ and HCHO columns, required for the comparison with the MAX-DOAS observations. To allow meaningful comparisons with the observations, the
model calculates daytime (8-16 local time) columns. More details on the model can be found in Stavrakou et al. (2015) and Bauwens et al. (2016).

Anthropogenic emissions over Africa are obtained from the Emission Database for Global Atmospheric Research (EDGAR4.2, edgar.jrc.ec.europa.eu) for 2008, and the NMVOC emissions are speciated according to the RETRO database (Schultz et al., 2007). Vegetation fire emissions are provided at $0.25° \times 0.25°$ resolution by the Global Fire Emissions Database version
4 (GFED4s), which accounts for the contribution of small fires based on active fire-detection data (Randerson et al., 2012; Giglio et al., 2013) and are distributed vertically according to Sofiev et al. (2013). Isoprene emissions are obtained from the MEGAN-MOHYCAN model (Müller et al., 2008; Stavrakou et al., 2014) for the study years at a resolution of $0.5° \times 0.5°$





(emissions.aeronomie.be). Based on these inventories over Africa, the anthropogenic emissions are estimated at 62 Tg CO, 1.3 Tg NOx-N and 23.3 Tg VOC, the biomass burning emissions at 152 Tg CO, 2.6 Tg NOx-N and 41 Tg VOC in 2013, and at 140 Tg CO, 2.4 Tg NOx-N and 38 Tg VOC in 2014, and the isoprene emissions are estimated at 79 Tg in 2013 and 90 Tg in 2014. The chemical oxidation mechanism of formaldehyde precursors is discussed in Stavrakou et al. (2009, 2015). The chemistry is

solved using the kinetic preprocessor (KPP, Damian et al., 2002).

The model simulates $NO_2$ and HCHO profiles and total columns at the station site for 2013, 2014, and 2015 (Fig. 7). The contribution of different emission sources to the total modelled HCHO column is calculated based on three model simulations ($CH_4$, BIO, and BIOANT) in which different sources are suppressed. Only HCHO formation from methane oxidation is included in the $CH_4$ simulation, whereas biogenic and anthropogenic sources are successively included in the BIO and BIOANT

simulations, respectively. In order to avoid feedbacks on the emissions on the concentrations of oxidants, monthly fields of OH, $HO_2$, NO, $NO_2$ and $NO_3$, obtained from a simulation including all source categories, are archived and used throughout the $CH_4$, BIO, and BIOANT simulations. For 2015 the $CH_4$, biogenic and anthropogenic contributions were taken from 2014, whereas biomass-burning climatological emission data (1997-2004 averages) were used for that year.

The IMAGES model results show that methane oxidation provides a significant background contribution to the modelled

columns at this location, accounting for about half of the total column outside the fire season, whereas circa 30% of the column is due to isoprene oxidation, and the remainder to anthropogenic sources. Emissions from vegetation fires lead to a strong column enhancement between June and September, and contribute to the total column by up to 40% in 2013 and up to 30% in 2014, as illustrated in Fig. 7. In Sect. 4.2.3 we compare the MAX-DOAS measurements to IMAGES model data and satellite retrievals.

**3.5   Satellite OMI and GOME-2 retrievals**

In this study, GOME-2 and OMI satellite retrievals are used, with respective local overpass times of 9h30 and 13h30, to compare in more detail the ground-based MAX-DOAS trace-gas measurements and satellite retrievals (see Sect. 4.2.3).

For the HCHO satellite data, we have used Level-2 formaldehyde products developed at BIRA-IASB and provided via the TEMIS website (http://h2co.aeronomy.be). A description of the products can be found in De Smedt et al. (2015). A common

mean altitude was assigned to all satellite pixels in an area of 100 km around Bujumbura. The satellite data selection criteria are as follows (De Smedt et al., 2015): residuals lower than three times the averaged fit residual, effective cloud fractions lower than 0.4, solar zenith angles lower than 70°, and individual vertical column errors lower than three times the column. For the tropospheric $NO_2$ data, GOME-2 TEMIS v2.30 (Boersma et al., 2004) and OMI TEMIS DOMINO v2.00 (Boersma et al., 2011) retrievals are considered. For both satellites, only cloud free pixels (cloud fraction $< 20\%$) within 50 km of Bujumbura

are selected, and daily mean values are calculated (Pinardi et al., 2014).

For a comparison of the ground-based and satellite data, we use monthly-mean cloud-free satellite averaging kernels (AVKs) to smooth the retrieved MAX-DOAS profiles. As the satellite AVK grid does not reach down to ground level, typically starting at a height around 1.5 km, the satellite height grid has been extrapolated and rebinned to the grid of the MAX-DOAS profile grid. For the OMI data, the pixels affected by the row anomaly have been excluded, and for $NO_2$ the provided (total) averaging





kernels have been transformed into tropospheric AKs, using the formula $AK_{tropo} = AK_{tot} * (AMF_{tot}/AMF_{tropo})$, following the Product Specification Document recommendations (http://www.temis.nl/docs/OMI_NO2_HE5_2.0_2011.pdf).

### 3.6 HYSPLIT

To investigate the transport of aerosols and trace-gases to Bujumbura (Sect. 4.2.2), we calculated air-mass backward tra-
jectories using the HYSPLIT model (HYbrid Single-Particle Lagrangian Integrated Trajectory model) from the NOAA Air
Resources Laboratory (Draxler and Hess, 1997, 1998; Draxler, 1999; Stein et al., 2015). We constructed an ensemble of 72 h
back-trajectories at 6 h intervals with the endpoints at retrieval heights of 750 m above ground level at the location of Bujum-
bura. The trajectories were then seasonally clustered for Dec 2014-Feb 2015, Mar 2015-May 2015, Jun 2015-Aug 2015, and
Sep 2015-Nov 2015, using the HYSPLIT Cluster Analysis Tool and NCAR/NCEP global reanalysis 2.5°meteorological fields.
This cluster analysis uses the total spatial variance to derive the number of clusters and a detailed description can be found on
https://ready.arl.noaa.gov/documents/Tutorial/html/traj_cluseqn.html.

## 4 Results and discussion

### 4.1 Seasonal and diurnal variations

We find a clear seasonal pattern in the AOD values (Fig. 8), with minimum values in April and November-December and
maximum values in February and July-August. The peak in JJA is the strongest, with monthly-averaged AOD values reaching
0.7, and individual measurements often reaching values well above AOD= 1. This variability is also reflected in the trace-gas
VCDs for HCHO, where we see a similar strong maximum in the JJA period, with a small secondary maximum in February.
For $NO_2$, the yearly variability is less pronounced, although we still see a minimum in MAM and maxima in JJA and DJF.
An overview of typical seasonal values for the aerosol optical depth, trace-gas vertical column densities, and near-surface
extinction coefficient and volume mixing ratios can be found in Table 3. The observed seasonality agrees well with the known
dry-wet seasonality, and related fire activities, in the Bujumbura region (see Sect. 2). The July-September maximum observed
in the aerosol and HCHO coincides with the peak of the local dry season and forest fires in JJA and 'small harvest' fire activities
in late SON, and the smaller peak in February with the fire activities related to the 'big harvest' season. The weak seasonality
for $NO_2$ might be explained by its short lifetime of about 1-2 hours, which limits the transport of $NO_x$ from fire-active regions
to the measurement site. To a large extend, measurements of $NO_2$ reflect the influence of local sources.

All MAX-DOAS retrievals show a clear daily pattern (Fig. 9), with a pronounced morning peak followed by a decrease in
the late morning and in the afternoon. As can be seen by the green numbers in the figure, the cloud-screening method removes
on average more data in the afternoon, however, this filtering does not affect the observed variability as presented in Fig. 9.
The only noticeable difference is that the spread in the aerosol 15-16 h AOD values becomes slightly more narrow after cloud
screening . For more information on the impact of the cloud screening, see the discussion in Sect. 4.1.1.





The aerosol optical depth peaks around 10-11 AM, followed by a minimum around 3 PM local time, and another rapid increase at 5 PM local time. This pattern is also seen in the trace-gas measurements, for which the peak is stronger and occurs earlier, around 9 AM, and the decrease towards the afternoon minimum starts earlier compared to the aerosols. For $NO_2$ we again see a secondary increase in the late afternoon, which is not seen in the HCHO measurements. This diurnal pattern is consistent with peaks in local anthropogenic activities such as traffic in the city center, which is strong between 7-9 AM and 5-7 PM. The strong diurnal variability of HCHO and its similarity with $NO_2$ however are surprising, as we expect anthropogenic emissions to play only a minor role for HCHO (see Fig. 7) and photochemical HCHO production due to methane and biogenic VOC oxidation is expected to maximise later during the day. Unidentified processes might be at play and/or anthropogenic sources might have a stronger impact and diurnal variability, e.g. due to the use of charcoal in cooking and domestic heating in the morning which tend to be the coldest part of the day.

We do not find strong evidence for a weekly pattern in the HCHO retrievals, but we do see a pronounced weekend decrease in the $NO_2$ vertical columns and a slight decrease for the aerosol retrievals (Fig. 9).

### 4.1.1 Comparison to AERONET AOD measurements

As can be seen in Figure 10, we find a good correlation between the AERONET and MAX-DOAS retrieved AOD values after cloud screening. When using the non-cloud-screened AOD values for both data sets, we find correlation coefficients $R = 0.61$ and $R = 0.57$ are found, for the 360 nm and 477 nm wavelengths respectively. After applying the cloud-screening on our MAX-DOAS retrievals only, correlation coefficients increase to $R = 0.83$ and $R = 0.80$. For the 477 nm AOD values, we do find a larger offset from the 1-1 relation, showing that the AERONET values are systematically higher than the MAX-DOAS data. This was already seen in other data sets (Clémer et al., 2010; Gielen et al., 2014), and could be due to the difference in air masses traced by both instruments, with the MAX-DOAS not capturing aerosols at high altitudes.

When using cloud-screened data for both AERONET and MAX-DOAS retrievals, we find an even bigger improvement is obtained, with values up to $R = 0.92$ and $R = 0.88$, for the 360 nm and 477 nm wavelengths respectively. The observed offset of the 477 nm results remains. Applying cloud screening on the MAX-DOAS data retains about 30% and 40% of the data for the 360 nm and 477 nm wavelengths respectively. If also the AERONET data is cloud screened, an additional 10% of data are removed.

In the AERONET data, the observed diurnal variability for the aerosol optical depth is less pronounced, and the strong rise in AOD after 16 h is also not observed (see Fig. 9). It is unclear where this discrepancy originates from. One explanation could be the different viewing directions of the MAX-DOAS and AERONET instruments, with the cimel being a sun-tracking device, and the MAX-DOAS having a fixed pointing direction. However, as the MAX-DOAS is pointed towards the west, the cimel and the MAX-DOAS have relatively similar viewing directions in the afternoon, where we see the biggest difference between the two instruments. The observed minimum at 16 h could also be related to the sun geometry and the western viewing direction of our instrument, as forward-sun MAX-DOAS measurements are known to be difficult,

Another reason could be the development of clouds in the western direction in the afternoon. Due to the tropical location and altitude of Lake Tanganyika, it has a very high evaporation rate, which follows a clear diurnal pattern. At the north side of the





lake, i.e. at Bujumbura, evaporation is highest during the day (Verburg and Hecky, 2003). This gives rise to the development of a low cloud layer above the lake in the afternoon, as can be seen in Figure 11. The cloud-screening statistics of our cloud screening indicate that more data are removed in the afternoon, with on average 30% of the AM data remaining versus 13% of the PM data. However, as these clouds often remain low, a fraction of clouds is missed by our cloud-screening method, as this method uses the 30° and zenith elevation angle. These 'missed' low clouds could affect the measurements, especially the MAX-DOAS at lower elevation angles. As the AERONET instrument is a sun-tracking instrument, these low clouds would only start affecting the measurements when the Sun gets low.

### 4.1.2 Vertical profiles

Figures 12 and 13 display the average seasonal, weekly and daily aerosol, $NO_2$, and HCHO profiles for the Bujumbura MAX-DOAS retrievals.

The aerosol extinction profiles show a strong increase in JJA, with lowest values found in MAM. In JJA and DJF, we see evidence for an uplifted layer of aerosols, peaking around 1.4 km, which is about 600 m above the station. Even though the extinction decreases at higher altitudes, a significant fraction is still found at heights above 3.5 km. However, due to the low DFS values of the retrievals and low sensitivity of the MAX-DOAS to higher layers, as evidenced by the averaging kernels (Sect. 3.1.1), the exact profile shape at altitudes above 2.5 km remains uncertain.

A different profile shape is seen for the $NO_2$ trace-gas concentration profiles, where we see a stronger exponential decrease towards higher altitudes. With the exception of MAM, the vertical profiles are very similar. The surface concentrations are found to be the highest in JJA and SON. The vertical profile for HCHO resembles the profile seen for the aerosol, with an uplifted layer peaking around 1.2 km, and highest concentrations during JJA and SON.

To further describe the retrieved profiles, we define a so-called 'characteristic height' or $H_x$, similar to Vlemmix et al. (2015), i.e. the height below which x% of the total partial tropospheric column density resides. This allows to study the variability in profile shape. Figure 12 shows the yearly variation of the $H_{75}, H_{50}$ and $H_{25}$ characteristic heights, using only measurements in a time period of 1 hour around local noon. We see that for the trace gases the $H_x$ values remain nearly constant throughout the year, whereas for the aerosols, higher profile altitudes are seen in MAM and SON.

The concentrations have strong diurnal variability, seen in the left plot of Figure 13, with for all retrieved species a large change in concentration throughout the day. This makes it difficult to investigate the variability in profile height. We therefore again use the characteristic heights to study the typical emission height. For the trace gases, we find the characteristic profile height stays stable throughout the day. However, for the aerosols, especially at 477 nm, we do see a clear increase in characteristic $H_{75}$ height in the late afternoon.

### 4.1.3 Comparison western-southern viewing direction

Since August 2015, the instrument has been measuring in alternating western and southern viewing directions. The western directions points towards the city and the lake, whereas the southern direction runs parallel to the lake, with little of the city in its line-of-sight. Due to an instrument malfunction, no southern measurements were taken in January 2016.





In Figure 14 we show the comparison between the retrievals for the two viewing directions. Markedly higher values are found for the southern viewing direction for the aerosol retrievals in the late SON and DJF; in August and September however, we find the opposite. For the trace gases slightly higher average values for the southern direction for the DJF months are obtained, but the difference is less pronounced than for the AODs. For August to October, we find slightly higher western values for $NO_2$. For HCHO, the southern measurements seem to be higher on average throughout the year.

Regarding the diurnal variation, we observe a similar afternoon decrease for the southern measurements, although it seems to be less pronounced, especially for the aerosols retrievals. This could indicate that the observed difference in the diurnal behaviour between the MAX-DOAS and cimel measurements, as discussed in Section 3.2, can be (partly) attributed to the cloud development above the lake or direct sunlight interfering with the measurements. A more detailed discussion on the agreement between the AERONET data and the southern measurements is difficult however given the very different viewing directions of both instruments. For HCHO, the southern and western values start to deviate more in the afternoon, which is not as pronounced for $NO_2$.

The correlation plots for the two viewing directions (Fig. 15), with the data rebinned in 1 h time intervals, reveal we have correlation coefficients ranging from $R = 0.72$ to $R = 0.93$ after applying the cloud screening. For the AOD retrievals $R > 0.80$ for both wavelength ranges are obtained, with western values on average higher than the southern. For the trace gases the cloud screening does not have as big an impact on the correlation, compared to the AODs. For $NO_2$ we find a correlation coefficient of $R = 0.72$, and again the western measurements result in slightly higher values. A very good correlation of $R = 0.93$ is seen for the HCHO VCDs with slightly higher southern values. These results indicate that the aerosol and $NO_2$ retrievals are much more influenced by the city emissions, compared to the HCHO measurements. This again suggests that the observed strong diurnal variability of the HCHO measurements is probably not (strongly) related to city-center anthropogenic emissions.

## 4.2 Impact of local and regional emissions

### 4.2.1 Comparison to MODIS fire data

In Figure 16 we compare the MODIS fire radiative power (Kaufman et al., 1998; Justice et al., 2002; Giglio et al., 2003), an indicator for fire activity, to the measured aerosol and trace-gas columns at the station. For this we choose a region around Bujumbura, for which our backtrajectory calculations (see Sect. 4.2.2) show they are the starting point of most air-mass trajectories reaching the station, i.e. the area between latitudes -15° and 4°, and longitudes 25° and 45°.

There is a clear correlation (see Fig. 16) between the monthly variability seen in the MODIS fire radiative power, and the MAX-DOAS aerosol optical depth and HCHO vertical columns, which shows that the MAX-DOAS is sensitive to biomass-burning activities from the surrounding regions. For the aerosols and HCHO, we see correlation coefficients between 0.58 and 0.63, where for $NO_2$ we only find values of 0.26. This can be partly understood by the difference in lifetime between $NO_2$ (a few hours) and the HCHO precursors (between several hours and several days). $NO_2$ is mostly affected by local emissions and less by regional transport from biomass-burning areas. Anthropogenic emissions are furthermore by far a larger contribution to the total $NO_x$ sources than to the precursors of HCHO.



### 4.2.2 Comparison to HYSPLIT back-trajectory models

Figure 17 presents the back-trajectory analysis calculated with the HYSPLIT model, for four different periods throughout the year. The air masses reaching Bujumbura originate predominantly from the east and south-east, with the exception of trajectories in the SON-DJF period, where the west and north-west also contribute. From the MODIS fire data (Fig. 6) we

know that these regions from which the back-trajectories originate are not heavily affected by biomass-burning activities, with the exception of the JJA period.

For the DJF period, the back-trajectories originate from an area relatively close to the station, between latitudes $[-6°, 2°]$ and longitudes $[22°, 38°]$. In this region there are some fire sources to the north of the equator, but the trajectories do not reach the extreme fire sources between $3°$ and $6°$ latitudes. In the MAM, most trajectories originate from the east coast, again away from

most fire sources. For JJA, the trajectories again originate from the east and south-east coastal regions, but we now find some strong fire sources in this area, which shows that for this period we expect an influence of fire emissions in our measurements, as is indeed the case (see Sect. 4). In SON many trajectories originate from the east, with additional trajectories coming in from the north and west. Especially around September, we see that there are still several strong fire sources in these regions, and strong contributions are expected in our measurements for this period.

### 4.2.3 Comparison to the IMAGES chemical transport model and satellite retrievals

In Figure 18 we compare the retrieved diurnal variation of the MAX-DOAS vertical-column densities of HCHO and $NO_2$ with calculated IMAGES VCD values. Due to computational restrictions, only diurnal IMAGES variability for 2013 was calculated. However, we do not expect large difference in diurnal variability for other years.

We find that for both gases, the IMAGES model underestimates the columns before local noon, and the strong diurnal

variation of the MAX-DOAS data is not seen. The afternoon values between both data sets agree relatively well. This could be (partly) due to the difference in horizontal resolution: the IMAGES model has a $2° \times 2.5°$ resolution, which corresponds to a rectangular area with sides about 220 km and 280 km around Bujumbura, whereas the MAX-DOAS measurements are more sensitive to emissions closes to the city center. The anthropogenic emission due to traffic and industry which are expected in the city center, could thus be underestimated by the model. The very peculiar orography of the region, with Bujumbura located

in a valley surrounded by mountains up to $2000 - 3000$ m further complicates the comparison with the IMAGES output.

Figure 19 compares the IMAGES results and MAX-DOAS measurements with the GOME-2 and OMI satellite products (see Sect. 3.5). The MAX-DOAS and IMAGES monthly-averaged profiles were smoothed with calculated satellite GOME2-B and OMI averaging kernels, in 2 hour intervals around the respective local overpass times of 9h30 and 13h30. When comparing satellite-retrieved vertical columns with MAX-DOAS-retrieved or IMAGES-derived trace-gas profiles, the use of the satellite

averaging kernel information allows to take into account the sensitivity profile of the satellite measurements in the comparison. By doing so, we remove the effect of the a-priori profile shape used in the satellite retrievals (Boersma et al., 2004). However, as the satellite AVKs are extrapolated to lower altitudes to allow the smoothing of the MAX-DOAS profiles, some uncertainty remains on the smoothed profile information closest to the ground.





For HCHO, the yearly morning variability seen in the MAX-DOAS vertical-column data is captured by the IMAGES model and satellite GOME-2 retrievals. We do however find higher MAX-DOAS VCD values for September-October, compared to IMAGES which shows a sharper drop in late JJA/early SON. For the afternoon values, the MAX-DOAS and satellite OMI data show similar variability, which is much less pronounced than for the morning values, although the large errors on the satellite

data make a detailed comparison difficult. The satellite retrievals show higher morning HCHO values compared to the afternoon, in agreement with our MAX-DOAS results, and already discussed in De Smedt et al. (2015). For both morning(GOME-2) and afternoon (OMI) data, there is a large disagreement between the MAX-DOAS data and IMAGES model in JJA, with the IMAGES model peaking much more strongly in early JJA, and the MAX-DOAS in late JJA/early SON. This deviation could be due to the IMAGES model 'missing' local emissions given the spatial sensitivity difference with the MAX-DOAS:

September-October are characterised by local emissions from agricultural fires from the small harvest season, to which the MAX-DOAS is more sensitive. A similar seasonality shift between the peak of the IMAGES and satellite HCHO emissions from early JJA to late JJA/early SON has already been observed for the larger South-African region and reported in studies such as Chevallier et al. (2009); Stavrakou et al. (2015); Bauwens et al. (2016).

For $NO_2$, the MAX-DOAS, satellite retrievals and the model seasonal variation agree well in the morning and relatively well

in the afternoon. For the GOME-2 morning data, the IMAGES model underestimates the ground-based and satellite retrievals, whereas in the OMI afternoon data, the model slightly overestimates the ground-based data in JJA and underestimates the measurements in the other seasons. We do find a large difference between the MAX-DOAS measurements and the OMI retrievals, with the satellite values almost twice as large between May-October. We need to be mindful of the large pixel size of the satellite retrievals (GOME-2: 40x80 km$^2$, OMI: 24x13 km$^2$), when comparing to MAX-DOAS measurements which cover

a much smaller region, typically a few tens of kilometers (Irie et al., 2011). However, given that $NO_2$ emissions are mostly related to anthropogenic activities, we would expect the MAX-DOAS to trace higher $NO_2$ concentrations than the satellite or the IMAGES model. As the $NO_2$ profile peaks near the surface, the uncertainty on the extrapolated satellite AVKs and the location in the valley could be (partly) responsible for the observed discrepancies. We also need to acknowledge that the observed$NO_2$ columns are small, reaching close to the satellite detection limit. This could cause retrieval biases, for example

due to an incorrect stratospheric correction.

## 5   Conclusions

We present over three years of ground-based MAX-DOAS measurements at the Central-African city of Bujumbura, which we compare to satellite measurements of trace-gas vertical columns and fire activity, IMAGES chemical-transport and HYSPLIT back-trajectory models. This is the first time such comparisons are performed in Central Africa. We find evidence of strong

anthropogenic, biogenic, and pyrogenic impacts on aerosols and trace gases $NO_2$ and HCHO. The MAX-DOAS proves well suited to retrieve aerosol optical depths and trace-gas vertical columns, together with information on the vertical distribution of these species.





First results show that the aerosol and HCHO seasonal variation, with a maximum in July-August-September and a secondary maximum in February, is driven by the alternation of rain and dry periods in the Bujumbura region. These dry seasons are associated with an intense signal associated to biomass burning, partly due to agricultural activities in the vicinity of Bujumbura and partly due to the transport of forest and savanna fire emissions from a region to the south and south-east of the city, as evidenced by the HYSPLIT back-trajectory models and the MODIS fire-radiative power satellite measurements. This strong seasonality is not seen in the $NO_2$ retrievals.

All MAX-DOAS retrievals show a clear daily pattern, with a strong peak in the morning and an afternoon minimum, followed by a sharp increase again in the late afternoon-early evening, most significantly for the aerosols and $NO_2$. This diurnal pattern is consistent with peaks in local anthropogenic activities in the city center. A weekend decrease is most strongly seen in the $NO_2$ VCDs, which again relates to the anthropogenic activities such as traffic and industry.

The weak seasonality but strong diurnal and weekly pattern in the $NO_2$ MAX-DOAS measurements shows that the observed $NO_2$ results mainly from local emissions close to the city center, in line with the short lifetime of this species (typically a few hours). This is further corroborated by the comparison of the measurements made in the viewing direction of the lake and the city center, i.e. to the west, and measurements parallel to the lake to the south. On average, the western values for the AOD and $NO_2$ VCD are higher, being more sensitive to anthropogenic emission in the city center, whereas HCHO seems to be less influenced by city-center emissions, with very similar VCD values for the southern and the western viewing directions. It remains unclear however, what the underlying cause is for the observed diurnal HCHO profile, typically indicative for anthropogenic emissions, given the fact that HCHO is less influenced by city-center emissions.

The retrieved aerosol optical depths show a very good agreement with co-located AERONET sun photometer measurements, especially after cloud-screening. However, the observed decrease in the afternoon followed by strong increase after 16 h, seen in the MAX-DOAS AODs is not found in the AERONET data. The origin of this discrepancy is yet unclear, but could be related to the development of afternoon clouds above lake Tanganyika. We indeed find that when measuring in the southern direction, parallel to the lake, the observed afternoon decrease-increase pattern in the MAX-DOAS measurements is less pronounced.

To further characterise the measurements we compare the MAX-DOAS trace-gas retrievals with the IMAGES chemical-transport model and GOME-2/OMI satellite observations. The IMAGES model does not reproduce the observed strong diurnal pattern seen in the MAX-DOAS data, which could indicate that the model is not sensitive to the anthropogenic city-center emissions, due to missing anthropogenic sources and/or grid resolution issues. To remove the influence of the a-priori profile in the satellite retrievals, we smoothed the MAX-DOAS and IMAGES monthly-averaged data with measured satellite GOME-2 and OMI averaging kernels, with respective local overpass times of 9h30 and 13h30. For both gases we find a relatively good agreement in the observed seasonality between ground-based, satellite and model data, especially for the GOME-2 morning retrievals. The OMI afternoon data show a larger discrepancy between the measurements and the IMAGES model, with the IMAGES model much more sharply peaked in JJA than the MAX-DOAS or OMI retrievals. It could also be related to a retrieval artefact at low relative azimuth angles, a most critical geometry for the MAX-DOAS.

However, due to the peculiar orography of the area and the higher sensitivity of the MAX-DOAS to local anthropogenic emission sources, a direct comparison to satellite and model retrievals with a coarser spatial resolution remains difficult.



The validation of the MAXDOAS observations of trace gases is further complicated by the lack of correlative ground-based observations like in-situ networks in this part of the world.

In future work we will expand the number of trace gases investigated at Bujumbura, including species such as glyoxal (CHOCHO), $SO_2$ and $BrO$. Additional information on emissions, such as from in-situ and mobile measurements could prove

invaluable in the interpretation and validation of our MAX-DOAS measurements.

*Acknowledgements.* This research was financially supported at IASB-BIRA by the Belgian Federal Science Policy Office, Brussels, through the AGACC-II project (SD/CS/07A) and the PRODEX project TROVA, and by the European Space Agency (ESA) through the GlobEmission DUE project (2011-2016). We acknowledge the use of data and imagery from LANCE FIRMS operated by the NASA/GSFC/Earth Science Data and Information System (ESDIS) with funding provided by NASA/HQ. BIRA-IASB HCHO scientific products from GOME-2 have

been jointly supported by Belgian PRODEX (A3C and TRACE-S5P) and EUMETSAT (CDOP-2). BIRA-IASB HCHO OMI developments are supported as part of the Sentinel-5 precursor TROPOMI level-2 project, funded by ESA and Belgian PRODEX (TRACE-S5P project). Multi-sensor HCHO developments are supported by EU FP7 (QA4ECV project). We acknowledge the free use of tropospheric $NO_2$ column data from the GOME-2 and OMI sensors from www.temis.nl. CIMEL Calibration was performed at the AERONET-EUROPE calibration center supported by ACTRIS (grant agreement 262254). The authors gratefully acknowledge the NOAA Air Resources Laboratory (ARL) for

the provision of the HYSPLIT transport and dispersion model and/or READY website (http://www.ready.noaa.gov) used in this publication. ERA-Interim data are provided courtesy of ECMWF.





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

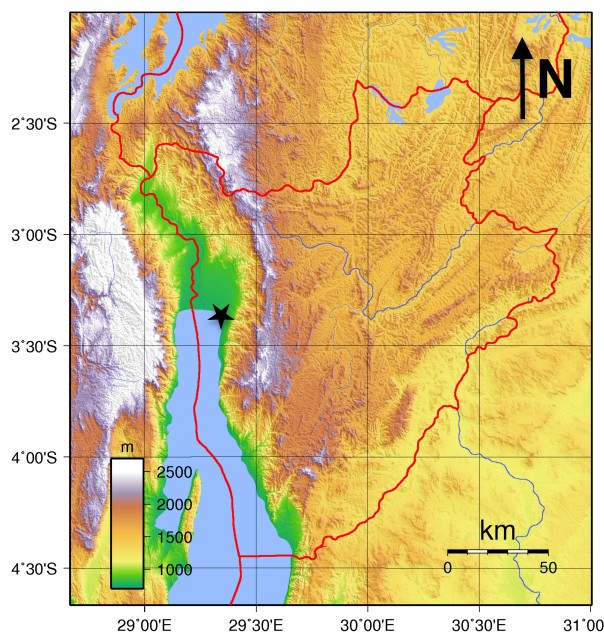

**Figure 1.** Topographic map of Burundi (source: http://en.wikipedia.org/wiki/Geography_of_Burundi). The location of Bujumbura is marked with a black star.





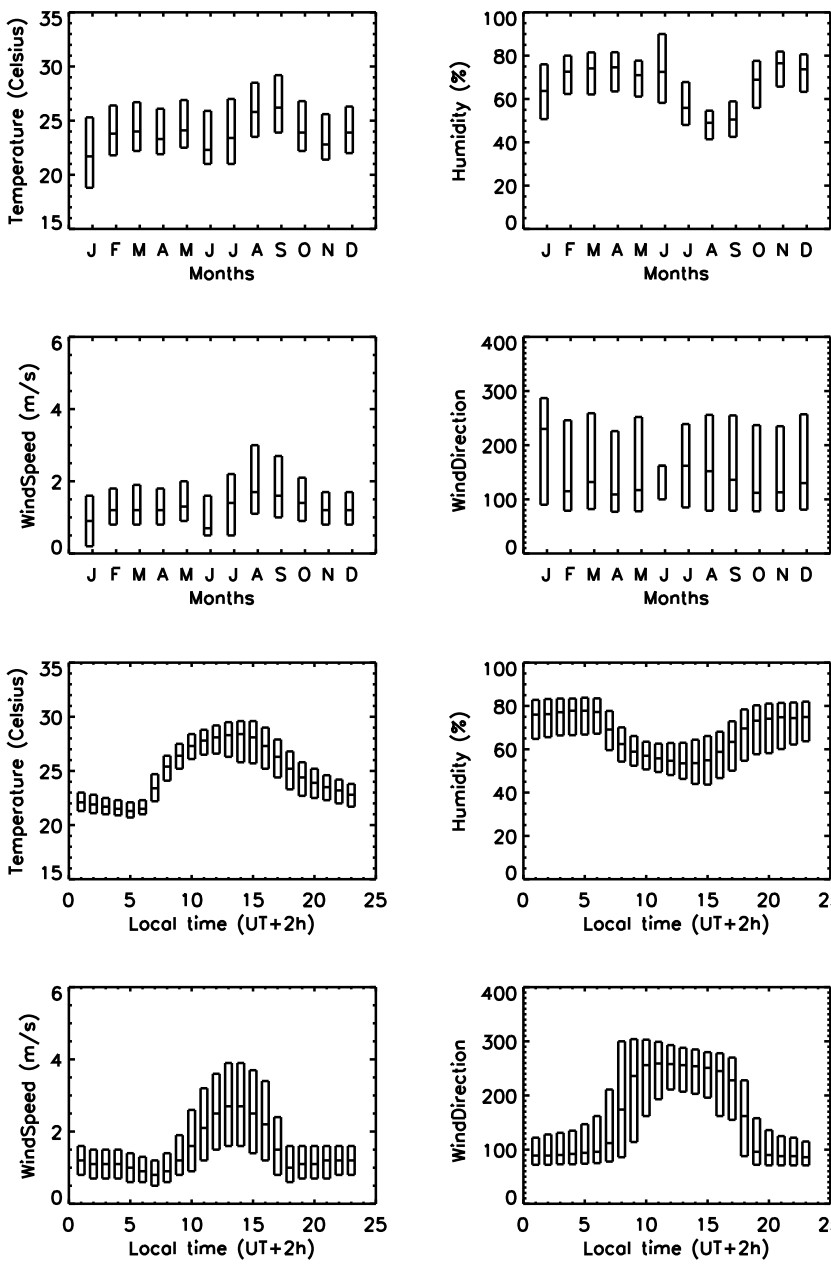

**Figure 2.** Seasonal and diurnal evolution of the temperature, humidity, wind speed, and wind direction (0° for winds from the north, 90° from the east) for Bujumbura in 2015. The upper four plots show the monthly averages with 25% and 75% percentile boxes. The lower four plots show the diurnal variation in local time.





**Figure 3.** The results of our bePRO radiative-transfer model fitting for an example day in 2015 (28/05/2015): the top left and right plots show the results for the aerosol modelling using $O_4$ at 360 and 477 nm respectively. The bottom left and right plots, the results for the trace-gas modelling of $NO_2$ and HCHO. For each plot the top box shows the the fit (full lines) to the measured DSCDs (crosses, in $10^{40}$ molec$^2$/cm$^5$ for $O_4$ and in molecules/cm$^2$ for the trace gases) at the different observational elevation angles. The second box gives the resulting aerosol optical depth or trace-gas vertical column density (in molecules/cm$^2$). The third and fourth box give the derived RMS (in percent) and DFS of the OEM retrieval.





**Figure 4.** Example of the bePRO aerosol at 360 and 477 nm (top left - top right) and trace-gas NO₂ and HCHO (bottom left - bottom right) retrieval for an example measurement (28/5/2015). The first box for each plot gives the averaging kernels, the second box the a-priori (red) and retrieved profile (black), and the third box the smoothing (green) and noise errors (blue).





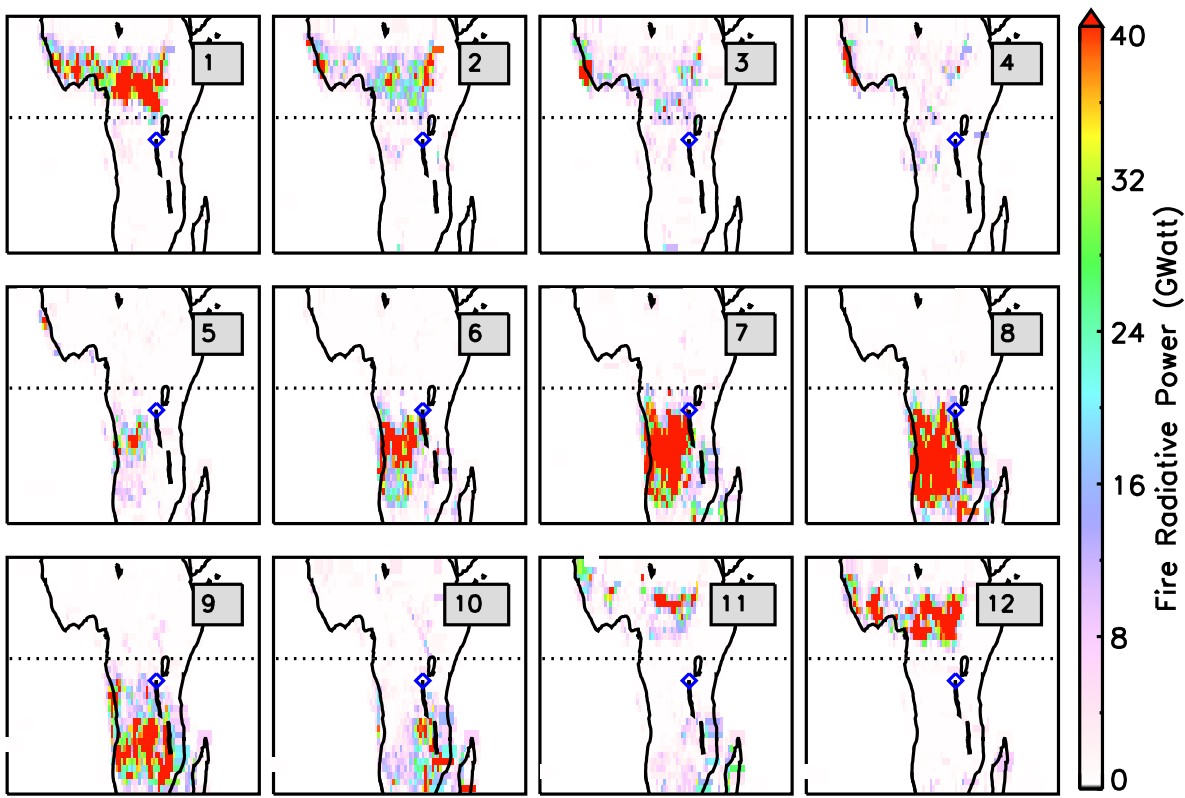

**Figure 5.** Monthly total MODIS fire radiative power (FRP) data for Central Africa for 2015. The location of Bujumbura is marked with a blue diamond, and the equator is represented with the dotted line.





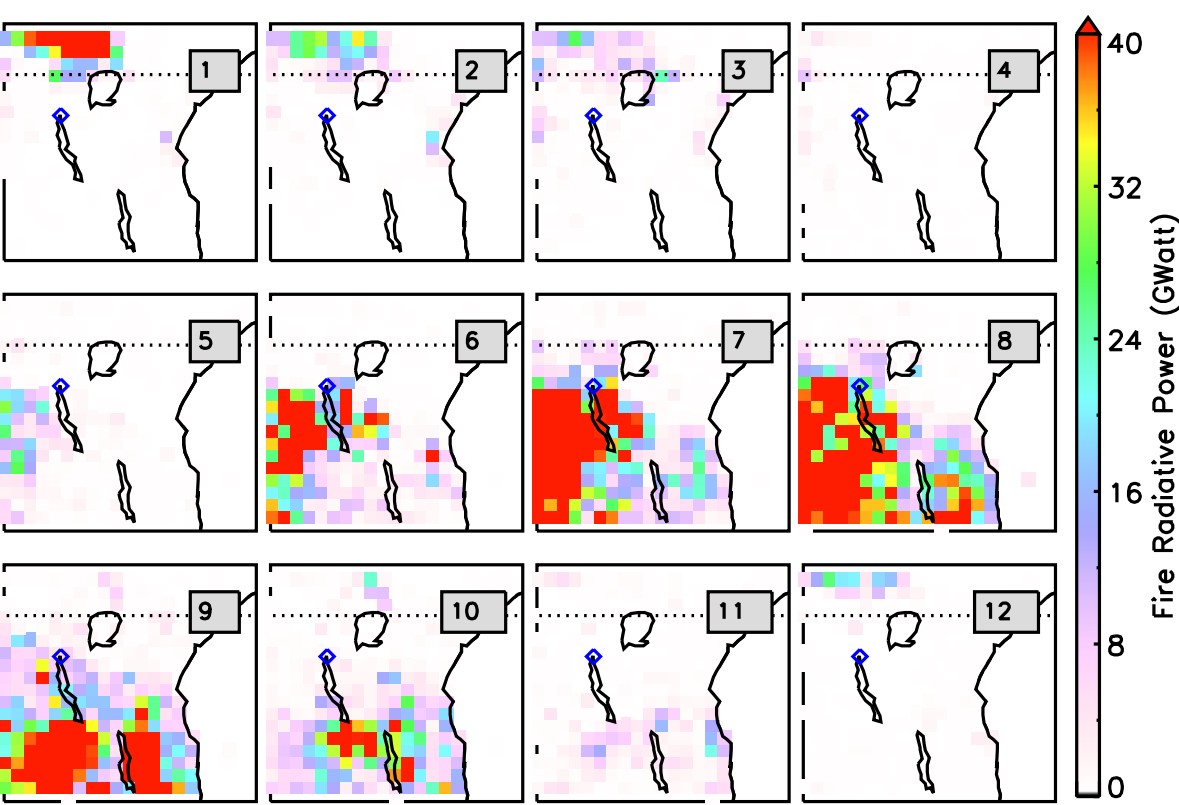

**Figure 6.** Same as Figure 5, but zoomed to the region between latitudes $[-15°, 4°]$ and longitudes $[25°, 45°]$.





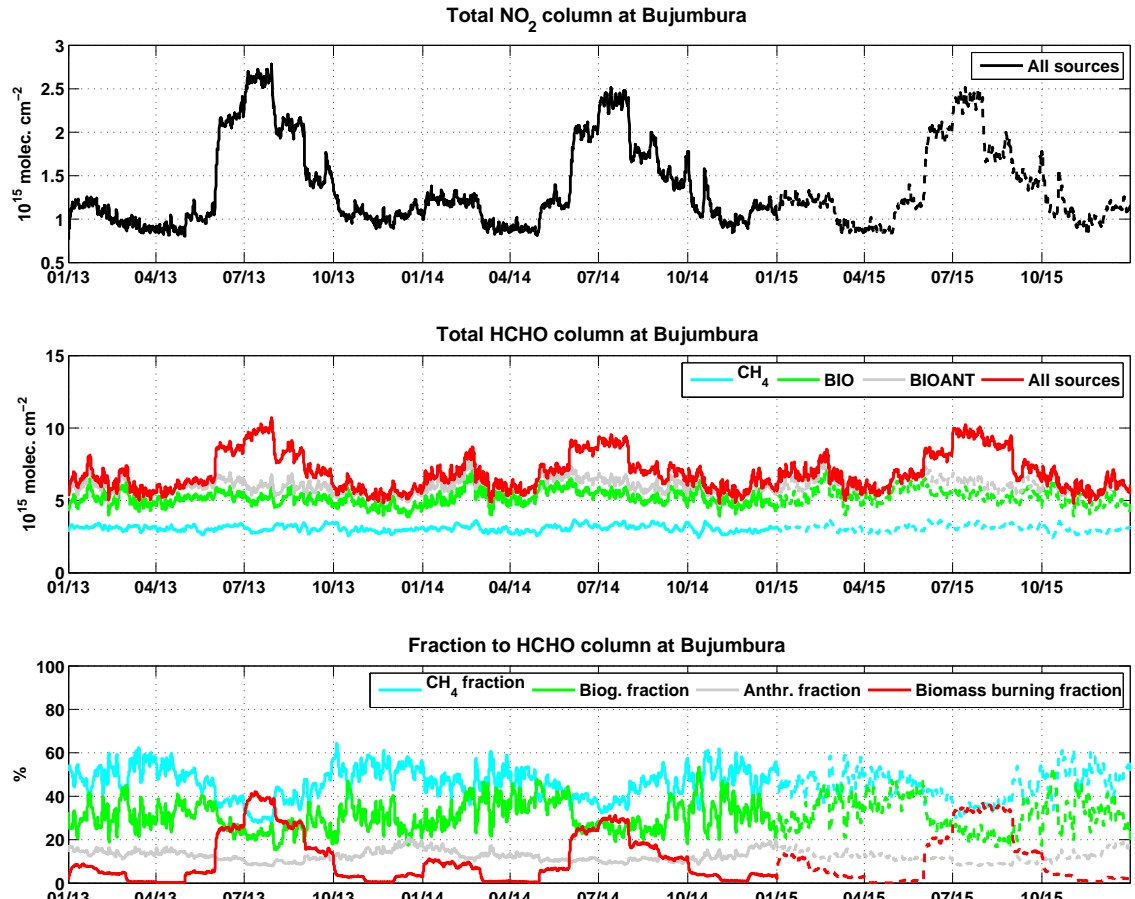

**Figure 7.** Top panel: Daily daytime $NO_2$ column (in $10^{15}$ molec cm$^{-2}$) at Bujumbura simulated with the IMAGES model. Middle panel: Daily daytime HCHO column (in $10^{15}$ molec cm$^{-2}$ simulated with the IMAGES model by accounting only for $CH_4$ oxidation (cyan), and by successively adding biogenic (green), anthropogenic (gray) and biomass burning (red) sources. Lower panel: Daily percentage fraction of the contribution of different source categories to the total HCHO column. The dotted lines in 2015 represent estimated emissions based on the 2014 measurements.





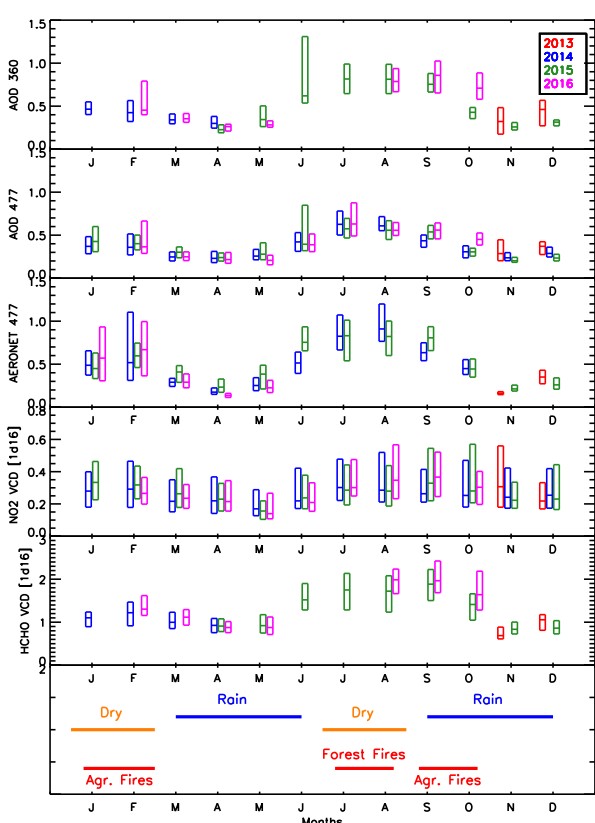

**Figure 8.** Yearly variability of aerosol optical depths and trace-gas vertical columns (in molecules/cm$^2$) at Bujumbura for respectively aerosols at 360 and 477 nm as measured with MAX-DOAS, AERONET AOD at 477 nm, and NO$_2$ and HCHO. The box indicates respectively the 25%-50%-75% percentiles. In the bottom panel we mark the main rain and dry seasons and fire activities in the Bujumbura area.





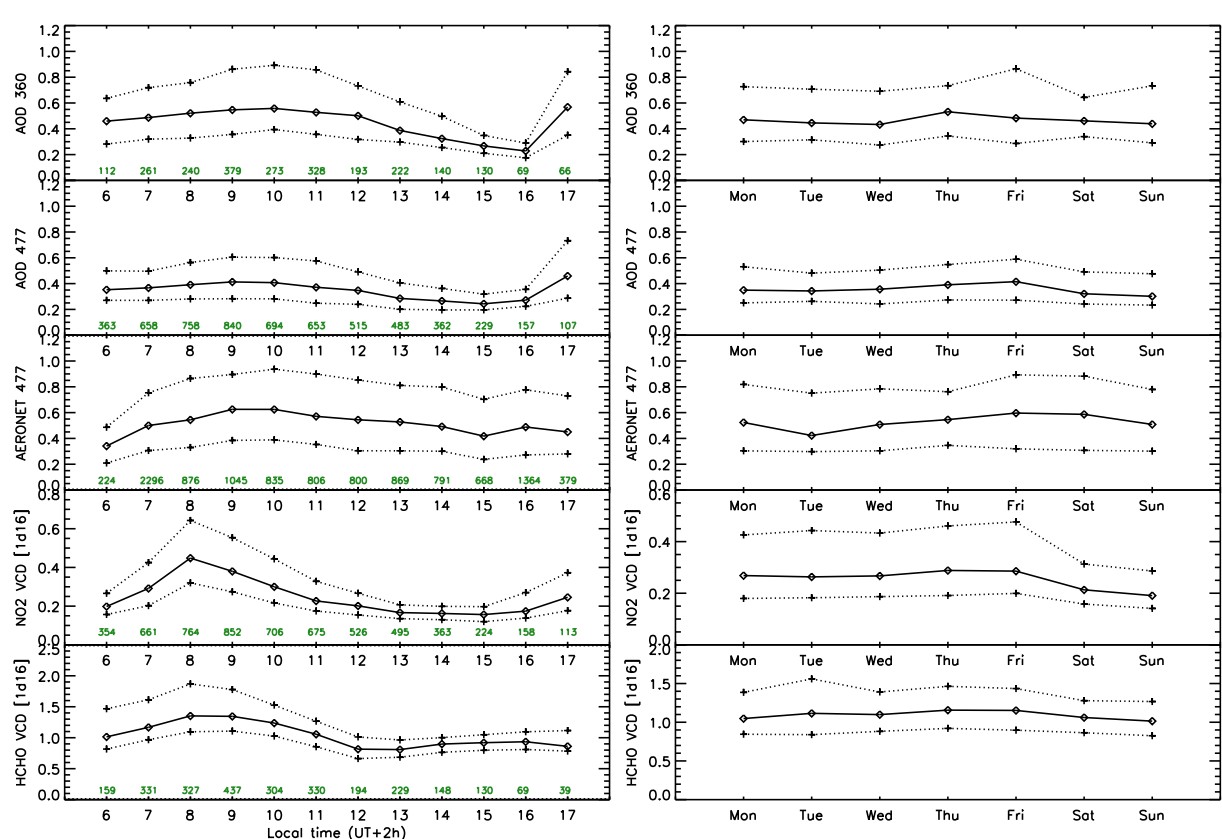

**Figure 9.** Diurnal (left) and weekly (right) variability of aerosol optical depths and trace-gas vertical columns (in molecules/cm$^2$) at Bujumbura for respectively aerosols at 360 and 477 nm as measured with MAX-DOAS, AERONET AOD at 477 nm, and NO$_2$ and HCHO. The lines give respectively the 25%-50%-75% percentiles. For the diurnal variability we also note in green the number of data points used for the calculation of the percentiles.





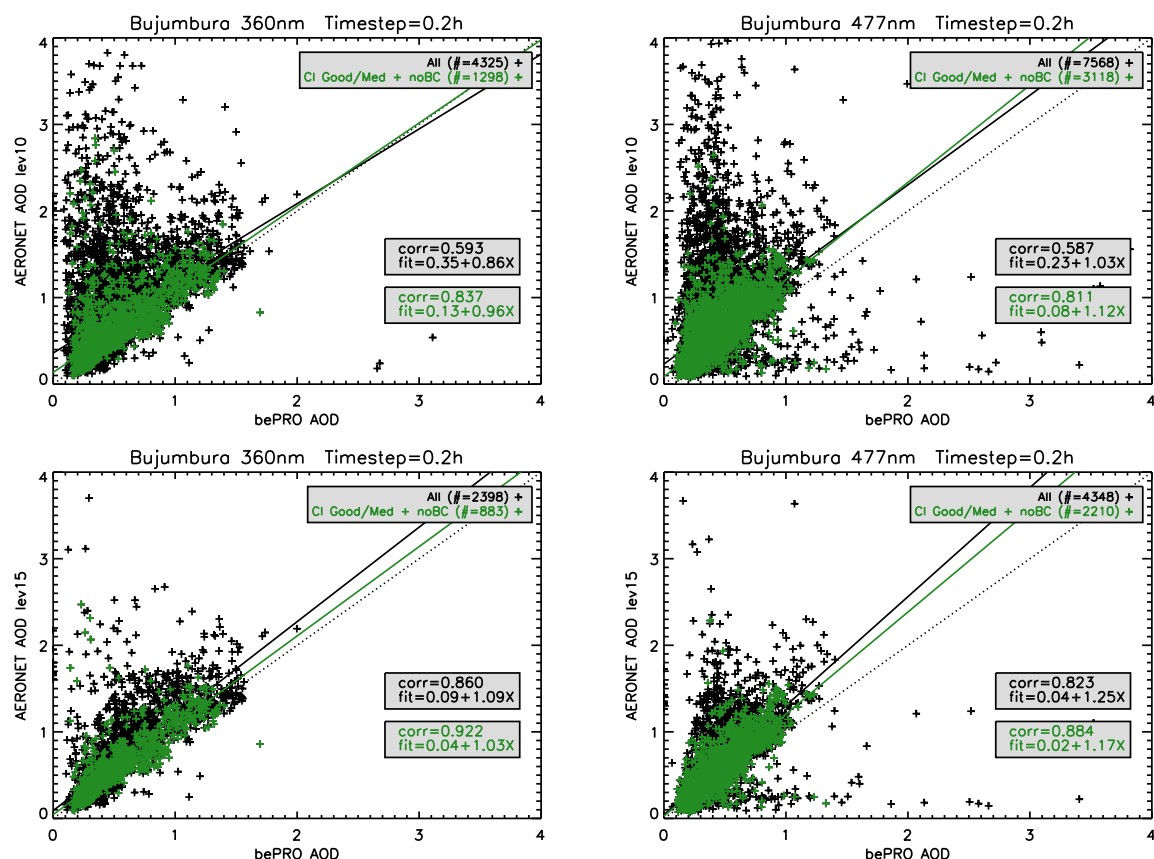

**Figure 10.** Correlation plots of bePRO MAX-DOAS AOD retrievals and measured AERONET AOD level 1.0 (upper plots) and level 1.5 (lower plots) data for Bujumbura at 360 (left) and 477 nm (right) in time steps of 0.2 h. The full non-cloud-screened MAX-DOAS data is given by black crosses. Cloud-screened MAX-DOAS data are marked in green crosses.





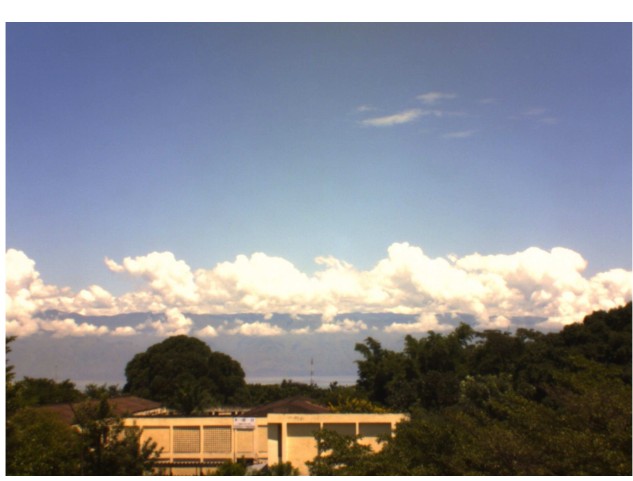

**Figure 11.** Development of a cloud layer above Lake Tanganyika at Bujumbura due to the lake's high evaporation rate. Picture taken with the web-cam installed at the measurement site, showing the viewing direction of the MAX-DOAS instrument.



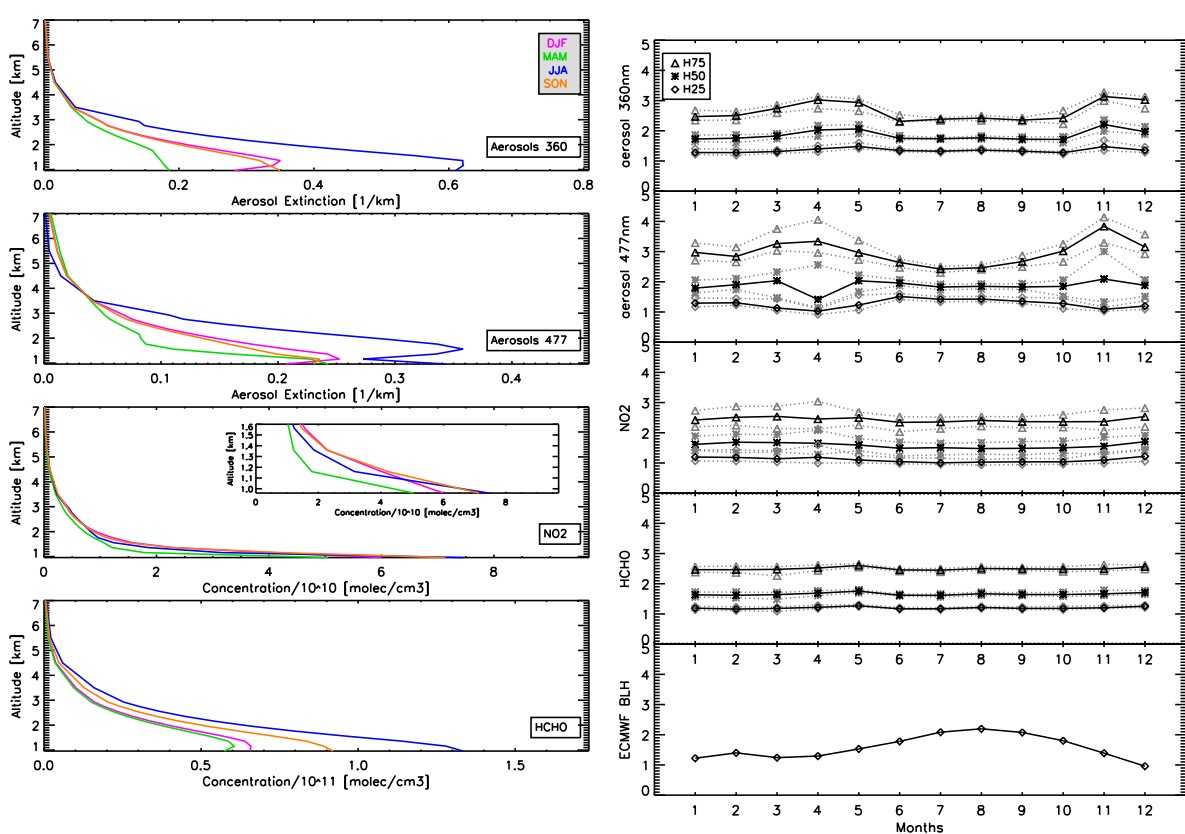

**Figure 12.** Left: Seasonal variability of aerosol and trace-gas profiles at Bujumbura for respectively aerosols at 360 and 477 nm, in aerosol extinction coefficient, and $NO_2$ and HCHO in concentration. Right: Seasonal variability of the 'characteristic heights' $H_{75}$ (triangles), $H_{50}$ (asterisks) and $H_{25}$ (diamonds) in km. The grey lines denote the 25% and 75% percentiles. The bottom panel shows the boundary layer height derived from ECMWF climatology.



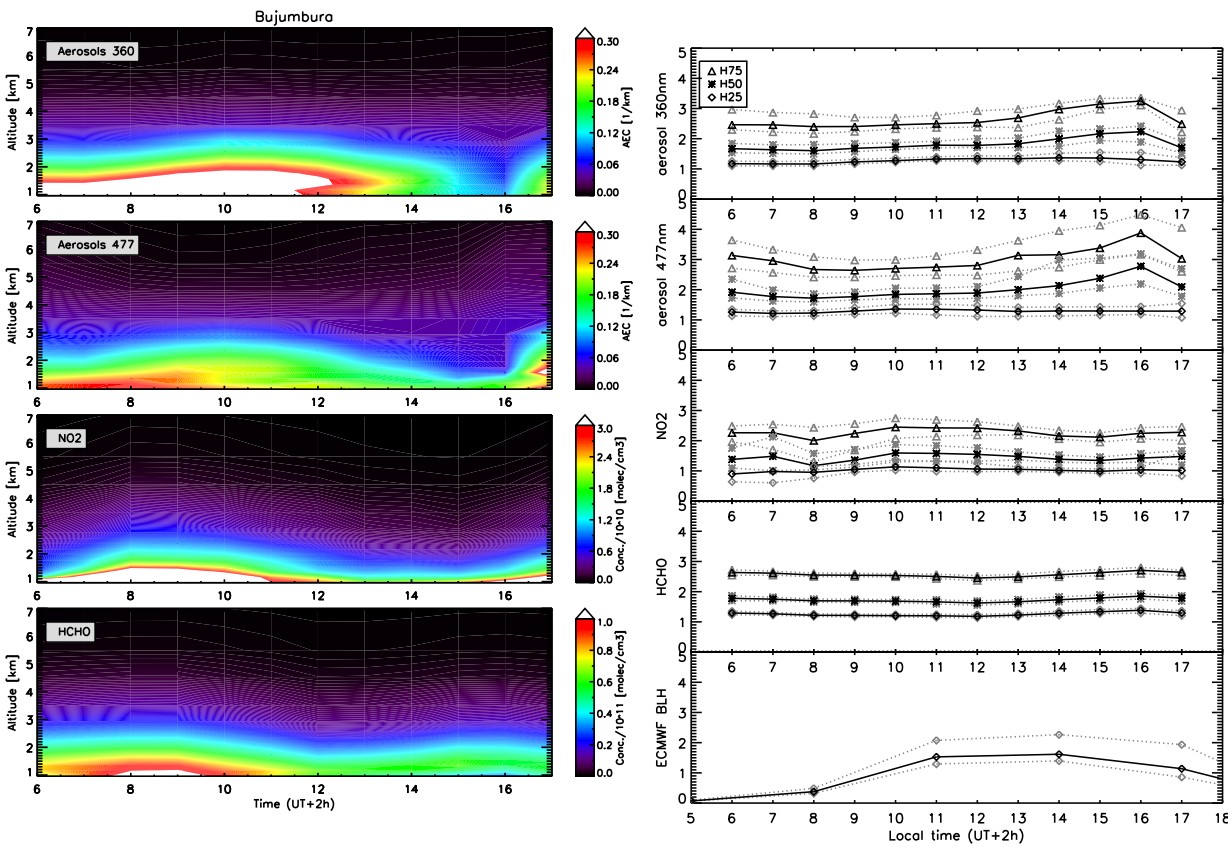

**Figure 13.** Left: Diurnal variability of aerosol and trace-gas profiles at Bujumbura for respectively the aerosol extinction coefficient at 360 and 477 nm, and the $NO_2$ and HCHO concentration. Right: Diurnal variability of the 'characteristic heights' $H_{75}$ (triangles), $H_{50}$ (asterisks), and $H_{25}$ (diamonds) in km. The grey lines denote the 25% and 75% percentiles. The bottom panel shows the boundary layer height derived from ECMWF climatology.





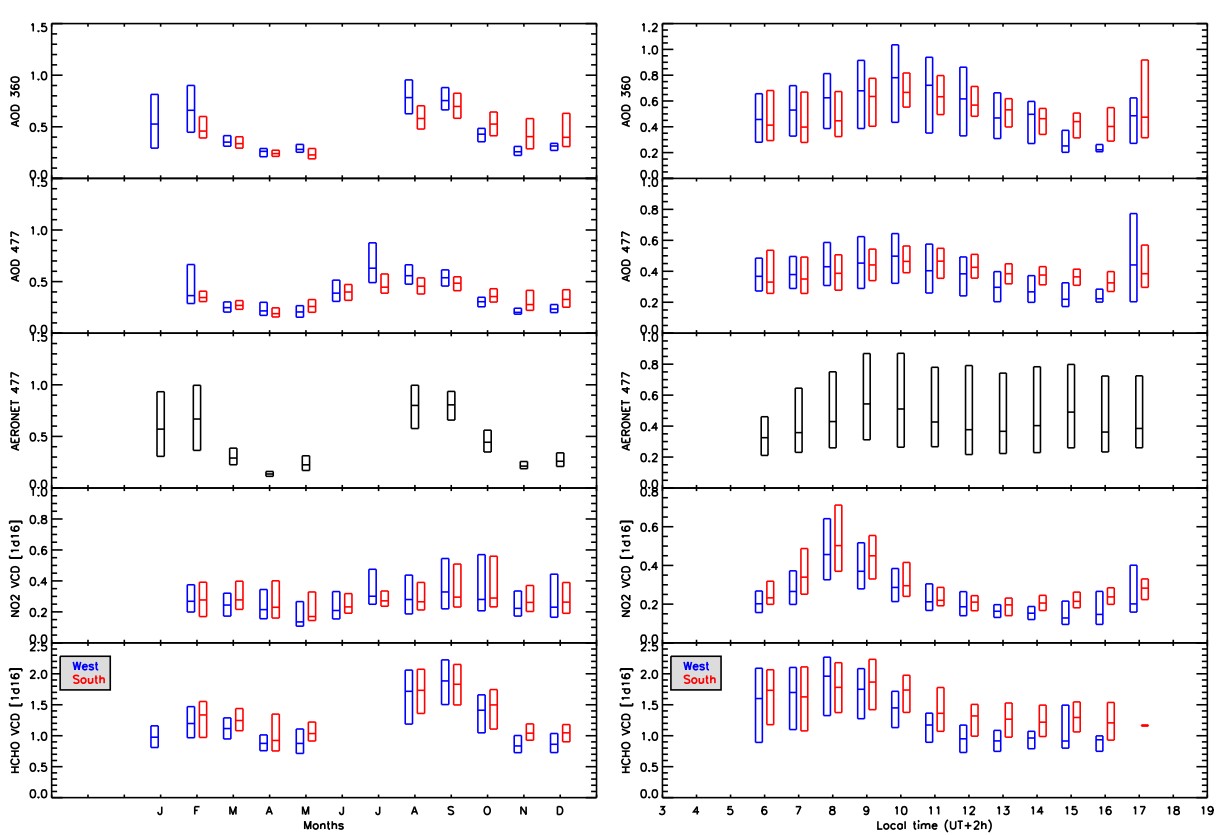

**Figure 14.** Monthly (left) and diurnal (right) variability of aerosol optical depths and trace-gas vertical columns (in molecules/cm$^2$) at Bujumbura for the western (city center/lake) and southern (rural) viewing direction, for respectively aerosols at 360 and 477 nm as measured with MAX-DOAS, AERONET AOD at 477 nm, and NO$_2$ and HCHO in from August 2015 onwards. The boxes give respectively the 25%-50%-75% percentiles.




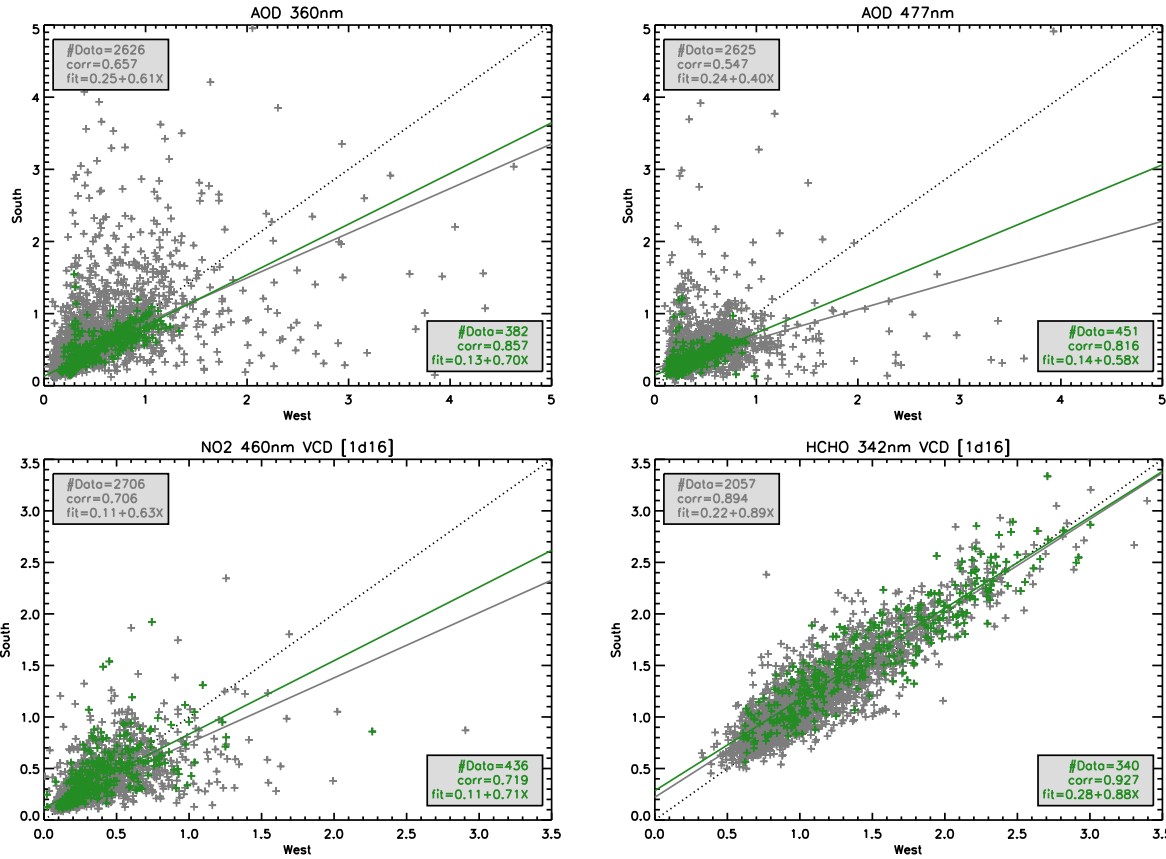

**Figure 15.** Correlation plot for the AOD at 360 and 477 nm, and $NO_2$ and HCHO VCD co-temporal retrievals (in $10^{16}$ molecules/cm$^2$) of the western (x-axis) and southern (y-axis) viewing directions. The grey crosses represent the full data set, the green are the cloud-screened measurements. All data were rebinned to 1 h time steps.





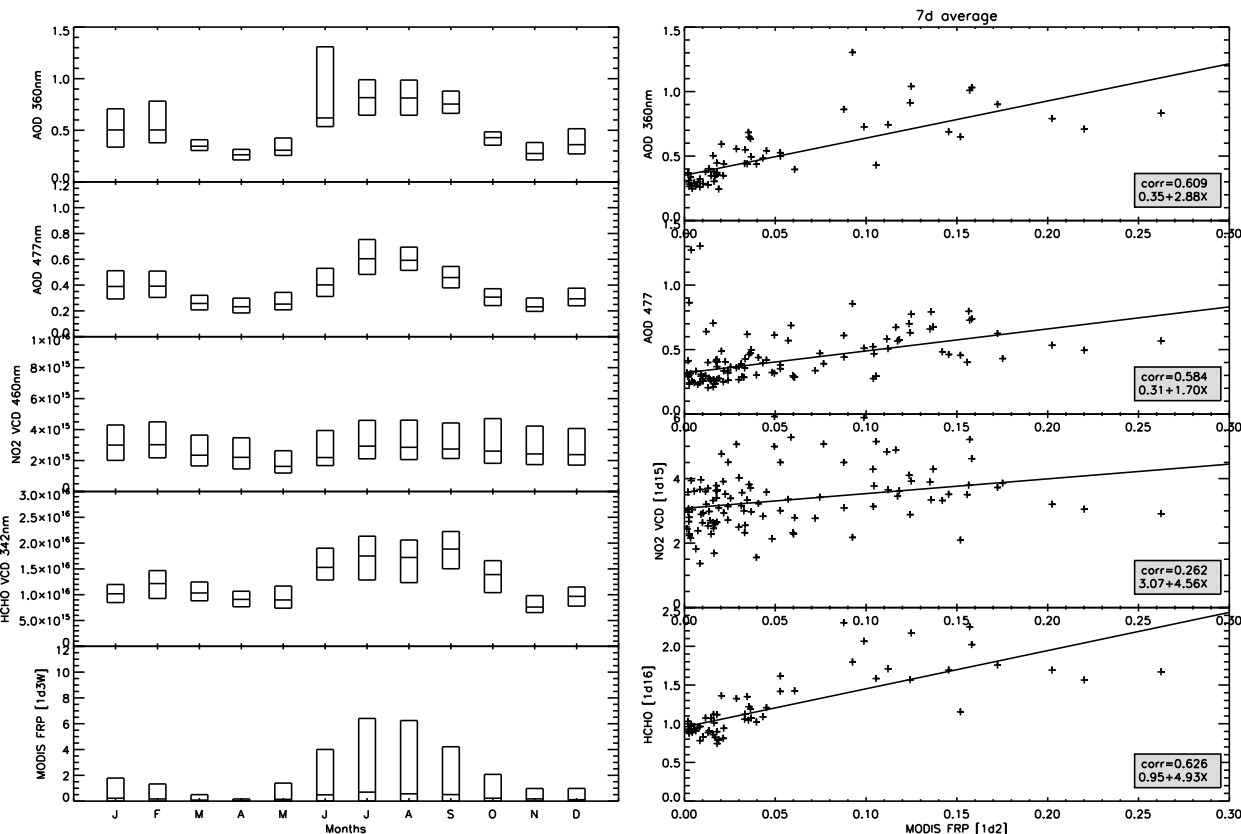

**Figure 16.** Left: Monthly average and 25-75% percentile bePRO AOD, and NO₂ and HCHO VCD and MODIS FRP values calculated in a box around Bujumbura with coordinates Lat1= $-15°$, Lat2= $4°$, Long1= $25°$, Long2= $45°$. Right: Correlation plots between retrieved bePRO AOD, NO₂ and HCHO VCD, and MODIS FRP (in the same box around Bujumbura), in bins of 7 days.



**Figure 17.** 72 h backward trajectories at the Bujumbura site at 750 m above ground level in 2015, for DJF (top left), MAM (top right), JJA (bottom left), and SON (bottom right). The upper part of each plot shows the relative fraction of the cluster to the total number of trajectories, the bottom part shows the height variation of each cluster in function of time.





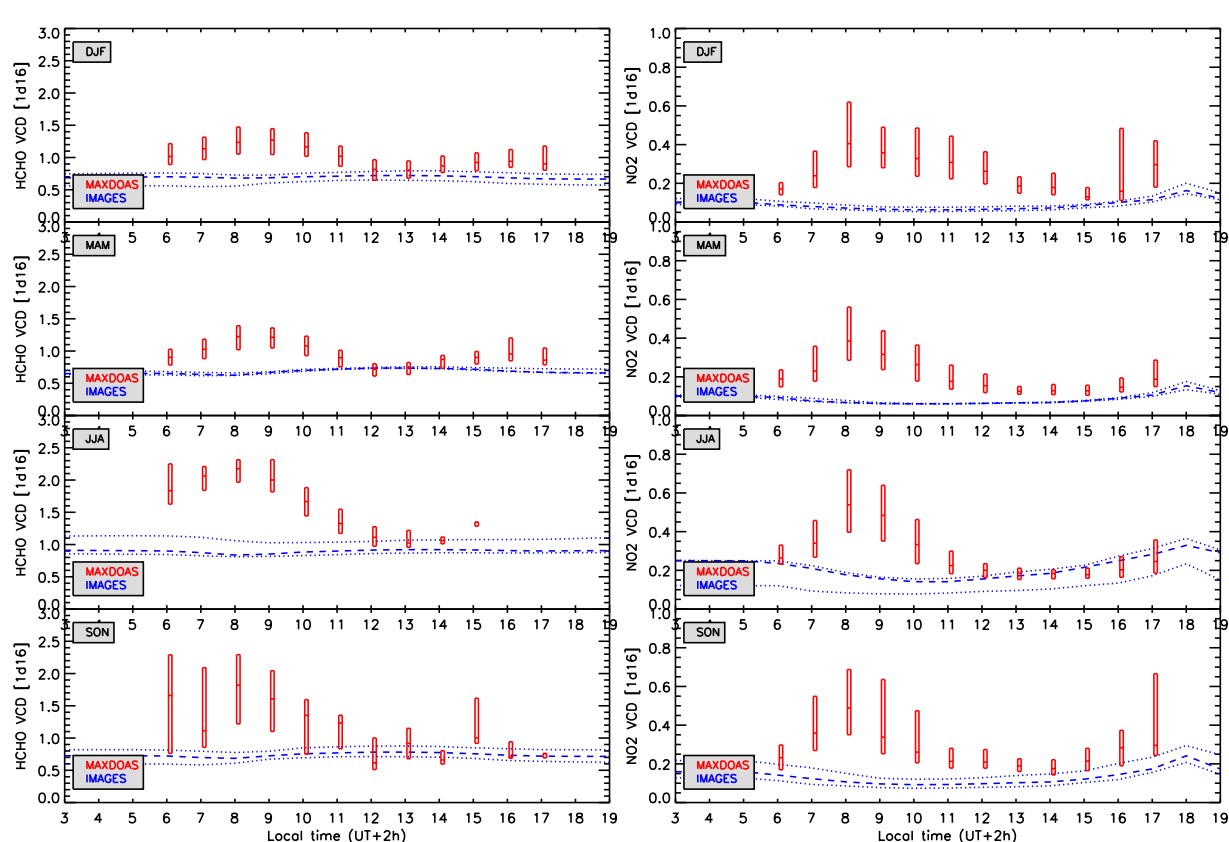

**Figure 18.** Comparison of diurnal MAX-DOAS measured VCDs (bars) for HCHO (left) and $NO_2$ (right) with the IMAGES model (lines) for 2013, separated for different seasons: DJF (December-January-February), MAM (March-April-May), JJA (June-July-August), SON (September-October-November). For both MAX-DOAS and IMAGES the 25%, 50%, and 75% percentiles are given.





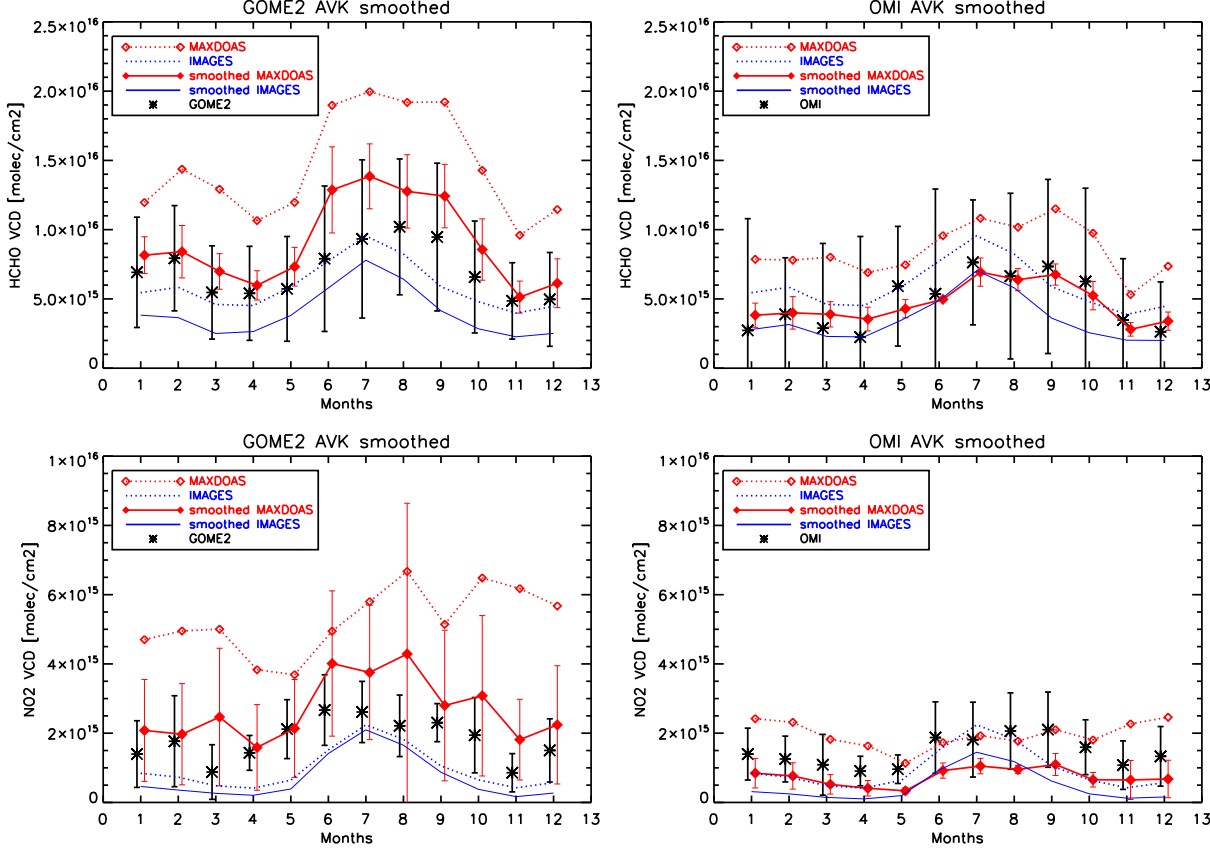

**Figure 19.** Comparison of monthly MAX-DOAS measured VCDs for HCHO (upper plots) and NO₂ (lower plots) with IMAGES model data and GOME2-B (left plots) and OMI (right plots) satellite measurements. The MAX-DOAS and IMAGES data are smoothed using the corresponding monthly-averaged satellite averaging kernels.



**Table 1.** DOAS settings for the slant-column retrieval of $O_4$ in the UV and the visible, $NO_2$ and HCHO.

| Parameter | Specifications |
|---|---|
| Wavelength calibration | Reference solar atlas, Chance and Kurucz (2010) |
| Intensity offset | Linear correction |
| Ring spectrum | Chance and Spurr (1997) |
| Closure term | 5th order polynomial |
| Reference spectrum | zenith spectrum of the scan |
| $O_4$ UV | |
| Fitting window UV | $338 - 370\,\mathrm{nm}$ |
| $O_4$ cross section | Hermans et al. (2003) |
| $O_3$ cross section | Bogumil et al. (2001), 243 K |
| HCHO cross section | Meller and Moortgat (2000), 297 K |
| BrO cross section | Fleischmann et al. (2004), 223 K |
| $NO_2$ cross section | Vandaele et al. (1998), 298 K |
| $O_4$ VIS | |
| Fitting window | $425 - 490\,\mathrm{nm}$ |
| $O_4$ cross section | Hermans et al. (2003) |
| $O_3$ cross section | Bogumil et al. (2001), 243 K and 293 K |
| $H_2O$ cross section | Harder and Brault (1997), 296 K |
| $NO_2$ cross section | Vandaele et al. (1998), 220 K and 298 K |
| HCHO | |
| Fitting window | $336 - 359\,\mathrm{nm}$ |
| $O_4$ cross section | Greenblatt et al. (1990) |
| HCHO cross section | Meller and Moortgat (2000), 293 K |
| $O_3$ cross section | Bogumil et al. (2003), 223 K and 243 K |
| BrO cross section | Fleischmann et al. (2004), 223 K |
| $NO_2$ cross section | Vandaele et al. (1998), 298 K |
| $NO_2$ | |
| Fitting window | $425 - 490\,\mathrm{nm}$ |
| $O_4$ cross section | Hermans et al. (2003) |
| $O_3$ cross section | Bogumil et al. (2001), 243 K and 293 K |
| $H_2O$ cross section | Harder and Brault (1997), 296 K |
| $NO_2$ cross section | Vandaele et al. (1998), 220 K and 298 K |



**Table 2.** Error budget of the retrieved aerosol optical depths and trace-gas vertical column densities. The total uncertainty is calculated by adding the different error terms in quadrature.

| Uncertainty related to (%) | AOD | NO$_2$ VCD | HCHO VCD |
|---|---|---|---|
| a-priori | 18 | 15 | 24 |
| smoothing and noise errors | 9 | 13 | 18 |
| cross sections | 5 | 3 | 9 |
| aerosol retrieval | | 5 | 4 |
| Total uncertainty | 21 | 21 | 31 |

**Table 3.** Median seasonal values for the aerosol optical depth (AOD) and near-surface extinction coefficient (AEC in 1/km) at 360 and 477 nm, and trace-gas vertical columns (VCD in $10^{15}$ molec cm$^{-3}$) and near-surface volume mixing ratios (VMR in ppb unit), as derived from the bePRO retrievals to the MAX-DOAS measurements. The near-surface extinction coefficient and volume mixing ratios were calculated by averaging the first two profile layers, i.e. at an averaged height of 200 m above the station.

| Season | Aerosol (360 nm) | | Aerosol (477 nm) | | NO$_2$ (460 nm) | | HCHO (342 nm) | |
|---|---|---|---|---|---|---|---|---|
| | AOD | AEC | AOD | AEC | VCD | VMR | VCD | VMR |
| MAM | 0.48 | 0.26 | 0.35 | 0.21 | 2.82 | 1.17 | 10.38 | 2.47 |
| JJA | 0.31 | 0.17 | 0.25 | 0.18 | 2.02 | 0.81 | 9.64 | 2.32 |
| SON | 0.81 | 0.52 | 0.55 | 0.26 | 2.70 | 1.17 | 17.18 | 4.57 |
| DJF | 0.45 | 0.24 | 0.34 | 0.20 | 2.63 | 1.21 | 11.77 | 3.16 |