# Peer review of "Characterisation of Central-African aerosol and trace-gas emissions based on MAX-DOAS measurements and model simulations over Bujumbura, Burundi."

_Atmospheric Chemistry and Physics, 2016_

## Referee Comment (RC1) · Anonymous Referee #1 · 13 Feb 2017

This paper presents a unique multi-year data set obtained over central Africa. Because there are almost no similar data records over this part of the globe, the data set presented here is of great value to obtain insights in tropical atmospheric chemistry and for validation of air quality models and satellite observations in this region.

The paper is very well written and presents many interesting and relevant findings in informative figures and tables. I recommend the paper for publication in ACP after the following comments have been addressed.

I have one main comment [C1], and four other comments [C2-C5]

[C1] Figures 9 and 13 show diurnal evolution of column amounts and profiles respectively. There are some aspects of both figures that raise questions:

First: in figure 9 a strong increase in AOD is seen at the end of the diurnal cycle for the MAX-DOAS product at 360 and 477nm, whereas a similar increase is not seen in the AERONET data.

Second: in figure 13 a strong variability is seen in the aerosol extinction profiles between morning and afternoon. This effect is not in line with the weak diurnal pattern in aerosol profile shapes (figure 13, right column) nor with the diurnal pattern in AERONET AOD. Even though the MAX-DOAS AOD shows a relatively weak diurnal variability in AOD when compared to the strong feature in fig. 13 upper left panel.

A possible explanation for this discrepancy could be that the MAX-DOAS results shown in these figures are affected by inhomogeneous temporal sampling over the seasons. This view is supported by the green numbers in figure 9. It is mentioned by the authors (p.9,l.27-28) that the diurnal patterns are not affected by the fact that less values are used to determine the afternoon percentiles. However, I am not convinced because it is quite probable that the results after 1PM, and especially the second half of the afternoon are dominated by measurements that are not representative for the entire year. This effect may also affect the diurnal cycles for NO2 and HCHO columns: although the column retrieval is generally quite robust compared to the profile retrieval, quality control (cloud filtering) may lead to an apparent diurnal cycle when - for instance - some seasons have considerably more cloudy afternoons than others. Perhaps it would make sense to be more strict in the selection of data. For instance: include only data from days where at least 3 out of 6 data points in the morning AND 3 out of 6 data points in the afternoon pass the quality control criteria. This ensures a more representative temporal sampling over the day and avoids the possibility that some hourly bins are dominated by one or two seasons, whereas others are representative for all seasons. Furthermore the temporal sampling should, in my view, be made consistent for the figures 9 and 13 and transparent in the sense that the reader should be able

to see how many data points from each season contribute to each bin of the diurnal cycle. Another possibility is to compute monthly averaged diurnal cycles and to give these equal weight (1/12) when determining the diurnal cycle averaged over one year. Please reconsider all interpretation throughout the manuscript of the patterns seen in these figures after the figures have been made again according to the suggestions above (or similarly).

Further comments:

[C2] Fig. 4. Please report relevant AOD in the caption of this figure.

[C3] Fig. 9 shows descending NO2 and HCHO columns after 8-9AM. It is argued that this diurnal variation is - at least for NO2 - due to peak emissions during the rush hour (p.10, l.4-10). It is indeed remarkable that the same is observed for HCHO, as noted by the authors, because of the different origins of this gas. The question therefore is: what are the possible explanations for the diurnal variability found for the HCHO columns. Before investigating further this mechanism in terms of emissions, transport and atmospheric photo-chemistry, alternative explanations - related to retrieval accuracy and further data analysis - should be considered. The issue of temporal sampling was already mentioned above: this could possibly affect this figure and should be checked first to my opinion. Secondly, I am not fully convinced that this diurnal cycle for HCHO is not related to variability in shielding strength by aerosols to trace gases above a certain altitude: for high AOD the MAX-DOAS measurements become at some point more and more insensitive to trace gases above a height of (say) 500-1000 meters (this altitude depends on the AOD). The fraction of the column for which this is the case may depend not only on the AOD, but also on the relative HCHO and aerosol profile shape and therefore vary throughout the day. In theory this effect should be taken into account by the MAX-DOAS profile retrieval method, but the effectiveness of this algorithm is doubtful, given the flat curves of Figure 13 (right column). I would like to know the authors view on this, and in case they share this concern ask them to add a few sentences on this topic to the discussion.

[C4] The aerosol profile shapes retrieved at 477 nm show in general little variablity throughout the seasons, but are most variable in months with low AOD (fig. 8). This could be an indication of several things: profile shapes are indeed more variable for conditions with low AOD, for instance because more sunlight reaches the earth surface leading to convection (a physical explanation); profile shapes just happen to be more variable in those months (i.e. coincidence of furthermore uncorrelated events); the profile retrieval is ineffective for conditions with high AOD (methodological limitation). I think this should be mentioned to the reader who is not familiar with the subtleties of MAX-DOAS profile retrieval and its interpretation.

[C5] Sect. 4.2.3 Although I think much effort is done to compare the satellite and the MAX-DOAS NO2 and HCHO columns, I think there ares still methodological limitations in addition to the point mentioned by the authors on p. 13, l.31-33. For instance: clear sky averaging kernels are used in relation to the satellite products, but these do not account for the presence of aerosols. This may be quite relevant in this region (depending on the season). In a sense, the comparison is only partially 'consistent' between satellite and MAX-DOAS (referring to the use of profile information to make a more valid comparison). Being aware of the practical challenges to add this element, I do not request to quantify this aspect in the study, but I would like to see this point mentioned in the discussion.

---

## Referee Comment (RC2) · Anonymous Referee #2 · 7 Mar 2017

The manuscript "Characterization of Central-African aerosol and trace-gas emissions based on MAX-DOAS measurements and model simulations over Bujumbura, Burundi." by Gielen et al. reports their 2-year MAX-DOAS observations of aerosol, $NO_2$, and HCHO at a site in Bujumbura, Burundi. The results are compared with co-located AERONET sun-photometer measurements as well as with satellite observations and 3D model simulations. Based on analyzing the entire data set in terms of temporal and spatial variations, the authors conclude that biomass-burning in surrounding regions of Bujumbura contributes significantly to the variation of aerosol and HCHO while $NO_2$ mainly originates from anthropogenic emissions in Bujumbura city center. Although

large, various, and valuable data are presented, I feel that the manuscript should be more focusing on analyzing characteristics of aerosol and trace-gas emissions than on explaining observation differences by different instruments. I would recommend the publication of the manuscript only if my following comments are well addressed.

**General comments**

From the title of the manuscript, I was expecting to see some detailed description on sources, possibly rates, and their variations (temporal, spatial) of aerosol and trace-gas emissions around the measurement sites. Especially, how is the yearly change of the emission, what could be the reason for that, and if possible what would be the atmospheric impacts? The authors spent large efforts on describing figures of measurement results but without much in-depth analysis. For example, Line 11-29 in Page 8, what is the reason for the difference in profiles measured in different seasons, why "the trace gases the $H_x$ values remain nearly constant throughout the year, whereas for the aerosols, higher profile altitudes are seen in MAM and SON"? Moreover, I do not get the idea that the comparison between sun-photometer/satellite and MAX-DOAS observations should be included in the manuscript, because they seems not quite related with characterizing emissions. In stead, I would suggest the authors to put more emphasis on analyzing the model and observation discrepancies, since ground observations are supposed to validate and improve model performance.

**Specific comments**

Line 9, Page 1: "biogenic emissions" → I did not find specific description on identifying biogenic emissions of aerosol and trace-gases in the manuscript.

Line 14, Page 1: "In contrast ... species (typically 1-2 hours)." → "In contrast, due to its short lifetime, $NO_2$ is seen to depend mainly on local emissions close to the city."

Line 15 − 28, Page 2: Readers of this paper are most likely familiar with atmospheric chemistry and know in general sources and sinks of aerosol, $NO_2$, and HCHO. I think what they might not know is the importance of characterizing emissions of these species in central Africa. What is the impact of African emission on the global scale?

Line 27, Page 3: Although the meteo. station was installed only from 2015, I think it is still necessary to use some historical data from nearby regions to illustrate that the meteorological condition is stable from year to year. Because changes in temperate will have strong influence on biogenic emissions and might lead to different HCHO and aerosol concentrations.

Line 9 − 10, Page 4: I suggest more description about the MAX-DOAS instrument, e.g., wavelength range, resolution, field of view, time for one duty scan, etc.. Or a reference should be mentioned.

Line 17 − 19, Page 4: What are the elevation angles?

Line 16 − 17, Page 5: Change to "... HCHO and $NO_2$ at 342 nm and 460 nm, respectively."

Line 25, Page 5: Can the temp. and press. profile of USSA represent the central African condition?

Section 4.1, Page 9 − 10: What about the year-to-year change of the observed AODs and VCDs? What could be the reason for that?

Line 6 − 7, Page 10: I estimate the life time of HCHO and $NO_2$ under overhead sun in summer in the central Africa is around 1.3 h and 1.7 h (@ OH = $1.5 \times 10^7 \, cm^{-3}$, $J_{\text{HCHO}} = 8 \times 10^{-5} \, s^{-1}$), respectively. Therefore, if the local anthropogenic emission can influence $NO_2$, it also works for HCHO unless this source does not emit HCHO.

Line 14 − 20, Page 10: Why the offset of the AOD regression between AERONET and MAX-DOAS only exists for 447 nm but not for 360 nm?

Line 21 − 22, Page 10: Delete "is obtained".

Section 4.1.2, Page 11: I noticed the aerosol profile below 2 km looks quite different between 360 nm and 470 nm in JJA months, i.e., there is a sharp decrease followed by an increase in 477 nm (Fig. 12). What is the reason for this?

Line 26, Page 11: I could not follow the conclusion that "This makes it difficult to investigate the variability in profile height". In principle, both photochemistry and boundary layer development influence the vertical distribution of $NO_2$ and HCHO. I think it is worthwhile to distinguish between these two factors at here. Probably with the help of aerosol profiles which to some extent can reflect the PBL change.

Line 11 − 12, Page 12: What could be the reason for the difference of diurnal variation between $NO_2$ and HCHO?

Line 13 − 20, Page 12: The observation at different viewing directions were performed in different years. Would this cause difference in $R$? And how to consider this possible year-to-year emission difference in identifying sources of $NO_2$ and HCHO? The same questions exist for Line 13 − 20 in this page.

Line 7, Page 13: I suggest to use the same geo-coordinate scale in Fig. 17, so that it is clearer for the statement "For the DJF period, the back-trajectories originate from an area relatively close to the station ...".

Section 4.2.3: If it is possible, I suggest the authors to reformat this section with more emphasis on the diagnostics of the model by MAX-DOAS and satellite observations. As illustrated in Figure 18 the model underestimates all measured species significantly, something must be wrong! How could the model performance been benefit from the measurements in this study?

Line 17, Page 13: I don't understand that why the authors did the model calculation for 2013 but not for 2014 or 2015 when they got most measurement results. This "computational restriction" even confused me when they already showed the model simulation for 2013 − 2015 in Fig. 7.

Line 19, Page 13: "... underestimate the columns ..." → Does the variability in Line 17 means columns for 2013?